# Aryl Hydrocarbon Receptor Activation by Benzo[*a*]pyrene Prevents Development of Septic Shock and Fatal Outcome in a Mouse Model of Systemic *Salmonella enterica* Infection

**DOI:** 10.3390/cells11040737

**Published:** 2022-02-20

**Authors:** Christiane Fueldner, Sina Riemschneider, Janine Haupt, Harald Jungnickel, Felix Schulze, Katharina Zoldan, Charlotte Esser, Sunna Hauschildt, Jens Knauer, Andreas Luch, Stefan Kalkhof, Jörg Lehmann

**Affiliations:** 1Department of Preclinical Development and Validation, Fraunhofer Institute for Cell Therapy and Immunology, 04103 Leipzig, Germany; christiane.fueldner@zv.uni-leipzig.de (C.F.); sina.riemschneider@izi.fraunhofer.de (S.R.); janine.haupt@izi.fraunhofer.de (J.H.); felix.schulze@uniklinikum-dresden.de (F.S.); katharina.zoldan@medizin.uni-leipzig.de (K.Z.); jens.knauer@izi.fraunhofer.de (J.K.); stefan.kalkhof@izi.fraunhofer.de (S.K.); 2Fraunhofer Cluster of Excellence Immune-Mediated Diseases CIMD, 04103 Leipzig, Germany; 3Department of Chemical and Product Safety, German Federal Institute for Risk Assessment (BfR), 10589 Berlin, Germany; harald.jungnickel@bfr.bund.de (H.J.); andreas.luch@bfr.bund.de (A.L.); 4Leibniz Research Institute for Environmental Medicine, 40225 Düsseldorf, Germany; charlotte.esser@uni-duesseldorf.de; 5Faculty of Life Sciences, University of Leipzig, 04103 Leipzig, Germany; shaus@server1.rz.uni-leipzig.de; 6Institute for Bioanalysis, Department of Applied Sciences, Coburg University of Applied Sciences, 96450 Coburg, Germany

**Keywords:** aryl hydrocarbon receptor (AhR), benzo[*a*]pyrene (BaP), immunomodulation, interferon-γ, interleukin (IL)-12, IL-10, IL-22, *Salmonella* infection, septic shock, systemic inflammatory response syndrome (SIRS)

## Abstract

This study focused on immunomodulatory effects of aryl hydrocarbon receptor (AhR) activation through benzo[*a*]pyrene (BaP) during systemic bacterial infection. Using a well-established mouse model of systemic *Salmonella enterica* (S.E.) infection, we studied the influence of BaP on the cellular and humoral immune response and the outcome of disease. BaP exposure significantly reduced mortality, which is mainly caused by septic shock. Surprisingly, the bacterial burden in BaP-exposed surviving mice was significantly higher compared to non-exposed mice. During the early phase of infection (days 1–3 post-infection (p.i.)), the transcription of proinflammatory factors (i.e., IL-12, IFN-γ, TNF-α, IL-1β, IL-6, IL-18) was induced faster under BaP exposure. Moreover, BaP supported the activity of antigen-presenting cells (i.e., CD64 (FcγRI), MHC II, NO radicals, phagocytosis) at the site of infection. However, early in infection, the anti-inflammatory cytokines IL-10 and IL-22 were also locally and systemically upregulated in BaP-exposed S.E.-infected mice. BaP-exposure resulted in long-term persistence of salmonellae up to day 90 p.i., which was accompanied by significantly elevated S.E.-specific antibody responses (i.e., IgG1, IgG2c). In summary, these data suggest that BaP-induced AhR activation is capable of preventing a fatal outcome of systemic S.E. infection, but may result in long-term bacterial persistence, which, in turn, may support the development of chronic inflammation.

## 1. Introduction

The aryl hydrocarbon receptor (AhR) is a ligand-activated transcription factor that is constitutively expressed in hepatocytes, epithelial cells in barrier tissues (i.e., gut, lung, skin) but also in different types of immune cells from the innate (i.e., macrophages, dendritic cells (DC), type-3 innate lymphoid cells (ILC3)) as well as the adaptive immune system (i.e., T helper (Th)17, Th22 cells) [1,2].

The receptor belongs to the Per-Arnt-Sim family of basic helix-loop-helix transcription factors, regulating detoxification, circadian rhythm and even cellular processes, such as differentiation and apoptosis ([3,4] and reviewed in [5]). Additionally, AhR plays a role in liver homeostasis, immune regulation and cell cycle control [6,7,8]. The signaling pathway of AhR is induced through the binding of planar and hydrophobic ligands. Representative endogenous ligands are, e.g., kynurenine and the tryptophan photoproduct 6-formylindolo[3,2-b]carbazole (FICZ) [9,10]. The most widely studied exogenous toxic ligands, 2,3,7,8-tetrachlorodibenzo-p-dioxin (TCDD) and benzo(a)pyrene (BaP), belong to the halogenated and polycyclic aromatic hydrocarbons, respectively [11]. The prototypical polycyclic aromatic hydrocarbon BaP is a ubiquitously present environmental pollutant. Exposure to high BaP concentrations may cause mutations and cancer [12,13]. BaP is produced by incomplete combustion of organic material; thus, it can be found in broiled food, cigarette smoke, industrial emissions and automobile exhaust [14].

While historically, the role of AhR was assumed in metabolizing and detoxifying drugs or xenobiotics through the induction of cytochrome P450 monooxygenases (CYPs) by xenobiotic AhR ligands, such as TCDD (reviewed in [15]), more recent reports have provided evidence that exposure to both xenobiotic (e.g., BaP) and dietary AhR ligands (e.g., indole-3-carbinole, I3C) can modulate innate and adaptive immunity. Thus, AhR activation may potentially result in enhanced susceptibility to infection or cancer and might trigger autoimmune disorders and allergies [16]. In regard to immunity to infection, it was observed that susceptibility and mortality caused by *Listeria monocytogenes* or LPS were significantly increased in AhR-deficient mice compared to wild-type mice [17,18]. The reason for the higher mortality rate of mice lacking AhR seems to be a hypersensitivity to LPS-induced septic shock, supporting the crucial role of AhR for a balanced systemic immune response to bacterial infection [18]. However, in murine models of pulmonary *Streptococcus pneumoniae* infection or systemic (i.p.) *Listeria monocytogenes* infection in the treatment of wild-type mice with AhR ligands, enhanced protective immunity was found against these bacterial pathogens, thereby causing higher survival rates, which could not be observed in *Ahr^−/−^* mice [17,19]. Apart from that, Moura-Alves and colleagues postulated that the AhR is not only an important regulator of the immune response but also represents a novel type of pattern recognition receptor (PRR). They found that certain bacteria express special pigments, representing a novel class of pathogen-associated molecular patterns (PAMPs), which can bind to the AhR, and thus, activate innate defense mechanisms against the invading bacterial pathogens [20]. This report is representative of others showing that bacterial metabolites are capable of modulating the host’s immune response, which in turn, may increase the tolerance of invading microorganisms. The plethora of those potential microbial AhR ligands, their different affinities and quantities, as well as their interaction with several known endogenous or xenobiotic AhR ligands make it difficult to create general concepts (reviewed in [21]). Therefore, any individual AhR ligand should be studied for its individual immunomodulating or even therapeutic potential.

Regarding all of these findings, it becomes clear that AhR plays a critical role during infection and seems to be important for a well-balanced immune response to invading pathogens. For this reason, it has to be elucidated that exposure to AhR-activating environmental pollutants, such as BaP, may significantly influence the course of infectious diseases. Therefore, this study was conducted to investigate the effects of environmentally relevant BaP concentrations [22,23,24] on the immune response against bacterial pathogens, utilizing a well-established mouse model of systemic *Salmonella enterica* infection, resembling features of the systemic inflammatory response syndrome (SIRS) in humans and endotoxin-induced septic shock.

## 2. Materials and Methods

### 2.1. Chemicals and Reagents

All chemicals or reagents were obtained from Sigma Aldrich (Taufkirchen, Germany) unless otherwise noted. Cell culture flasks and plates were purchased from Greiner Bio-One (Frickenhausen, Germany). RPMI 1640 culture medium (Biochrom, Berlin, Germany) was supplemented with 10 mM HEPES buffer, 2 mM L-glutamine, 10% (*v/v*) fetal calf serum (FCS) and 100 U/mL penicillin, 100 µg/mL streptomycin and 50 μM β-mercaptoethanol to be used as complete cell culture medium. *Salmonella enterica* growth experiments were carried out using Luria Bertani (LB) fluid medium or agar (both from Becton Dickinson, Heidelberg, Germany) or selective Xylose-Lysine-Desoxicholate (XLD) agar (Heipha Dr. Müller, Eppelheim, Germany). For in vivo experiments, the AhR ligands BaP and FICZ (Enzo Life Science, Lörrach, Germany) were dissolved in corn oil. For Griess reagent, 0.1% *N*-(1 naphthyl)-ethylenediamine-dihydrochloride was solved in ethanol (>99.8%) and mixed with a solution of 1% sulfanilamide solved in 5% phosphoric acid.

### 2.2. Mice

Female wild-type (WT) C57BL/6JRj mice were originally purchased from Janvier Labs (St. Berthevin Cedex, France). C57BL/6 AhR knockout (*Ahr*^−/−^) mice [25] were originally purchased from Jackson Laboratory (Bar Harbor, USA) and back-crossed in-house to WT C57BL/6JRj mice several times, while the *Ahr* deletion was confirmed by genotyping via PCR. Animals were used at 8–10 weeks of age for the experiments. Mice for septic shock experiments were housed as five or six animals per cage in the animal care facility of the Max Planck Institute for Evolutionary Anthropology (Leipzig, Germany) in a temperature- and humidity-controlled room (23 °C, 50% humidity) under specific pathogen-free conditions with 12 h/12 h of light/dark cycle and free access to pelleted standard rodent chow and water ad libitum. All experiments involving laboratory animals were conducted according to the European Communities Council Directive (86/609/EEC) and were approved by local authorities (registration no. TVV 05/09, TVV 57/15; Landesdirektion Sachsen, Leipzig, Germany). For organ collection, all animals were sacrificed using flow-controlled carbon dioxide (1 L/min). All efforts were made to minimize suffering of the animals.

### 2.3. Exposure to AhR Ligands In Vivo

To study the immunomodulatory potential of BaP in a standardized bacterial infection model, C57BL/6 mice were treated with 0.02 µg/kg or 2 µg/kg body weight (bw) BaP dissolved in corn oil or with corn oil alone (vehicle control). Previously, both BaP concentrations applied in this study were shown to be subtoxic in vitro and in vivo in C57BL/6 mice (Lehmann et al., submitted). BaP or corn oil was intraperitoneally (i.p.) injected two days prior to infection. After inoculation of bacteria, BaP was injected i.p. every third day.

For comparison and evaluation of the BaP-induced effects, we administered the high-affinity endogenous AhR ligand FICZ (1.25 µg/kg body weight).

### 2.4. Infection with Salmonella enterica Serovar Enteritidis

Two days after the first application of BaP/FICZ, mice were infected i.p. with 5 × 10^7^ CFU of the attenuated (LD_50_ = 3 × 10^7^ CFU) vaccine strain *Salmonella enterica* subsp. *enterica* serovar Enteritidis (S.E.; ade-/his-; SALMOVAC^®^SE, IDT Biologika, Dessau, Germany) [26,27]. Using this attenuated *S. enterica* strain, we previously developed a murine infection model for basic research of immunity to *Salmonella* and preclinical validation and immunological characterization of the vaccine prior to the application in poultry and swine [28,29]. In the present study, this infection model was utilized to evaluate the immunotoxicological impact of BaP during bacterial infection.

### 2.5. Extra- and Intracellular Bacterial Burden

For determination of total bacterial organ burden in the spleen and liver, the weight of each organ was determined. Then, a piece of spleen and liver was cut, weighed and homogenized in cold PBS using T 10 basic Ultra-Turrax^®^ (IKA^®^-Werke, Staufen, Germany). Subsequently, spleen and liver homogenates were diluted 1:10 in PBS (*w/v*). Log_10_-serial dilutions of the homogenates in PBS were plated onto selective Xylose-Lysine-Desoxicholate (XLD; Merck, Darmstadt, Germany) agar. Growing bacterial colonies were counted after 24- and 48-h incubation periods at 37 °C. Total bacterial burden in the spleen and liver was calculated in relation to the organ weight (CFU/g liver and CFU/g spleen).

Intracellular *Salmonella* burden in phagocytes from the spleen were determined as described previously [28]. In brief, 10^7^ splenocytes were plated onto XLD agar and shortly air-dried within a level-2 biosafety cabinet. Splenocyte-associated bacterial colonies delivered from phagocytes, which died under these conditions, were counted after 24- and 48-h incubation periods at 37 °C.

### 2.6. Detection of BaP and Its Metabolites in Liver Homogenates by Flow Injection Analysis MS/MS

Livers were isolated under aseptic conditions, weighed and cut into small pieces of tissue. Tissue pieces with defined weight were shock-frozen in liquid nitrogen and stored at −80 °C until analysis. The yield of BaP and the most prominent BaP-derived metabolites were detected by mass spectrometry, as previously described [30]. In brief, small tissue pieces from mouse livers were dissolved and homogenized in 1 mL ethyl acetate. For quantification, internal standards were added. The ethyl acetate phase was evaporated using a conventional vacuum centrifuge and resuspended in 50 μL of methanol/water mix (50:50 *v/v*). For analysis, 10 μL homogenate per sample were injected into a Shimadzu LC-20AD prominence (Shimadzu, Duisburg, Germany) HPLC system, online coupled to an API 4000 Q TRAP mass spectrometer (AB Sciex Instruments, Applied Biosystems, Darmstadt, Germany). Chromatography was performed on an Envirosep PP column (125 mm × 2 mm, 5 μm particle size, Phenomenex, Aschaffenburg, Germany) at 20 °C with a flow rate of 0.2 mL/min. HPLC separation was achieved by running a 23 min linear gradient with 2–20% B (with solvent A, 5 mM ammonium acetate in methanol, and B, 5 mM ammonium acetate in methanol/water mix (50:50 *v/v*)). For each analyte, the retention time and fragmentation patterns were obtained from reference standards (purchased from NCI Chemical Carcinogen Reference Standard Repository, Midwest Research Institute, Kansas City, MO, USA). Analytes were measured in scheduled MRM mode using negative APCI ionization (source temperature 350 °C). The two most intense ion transition pairs were used for quantification in relation to the corresponding internal standard. All concentrations were normalized to 1 g liver tissue.

Targeted analysis of BaP metabolites was performed by flow injection analysis MS/MS on an Agilent 1100 series binary HPLC system (Agilent Technologies, Waldbronn, Germany) coupled with a 4000 QTRAP linear ion trap mass spectrometer (AB Sciex, Concord, ON, Canada) equipped with a TurboIon spray source (source temperature: 200 °C). Flow injection analysis MS/MS (FIA−MS/MS) was performed with MS running solvent at ambient temperature. Quantification was achieved by positive and negative multiple reaction monitoring (MRM) detection in combination with the use of stable isotope-labeled and other internal standards. Data evaluation for quantification of metabolite concentrations was performed with the MetIQTM software package. Metabolites with ratios above 1.5 or below 0.67 are assumed to be up- or down-regulated. Significance was computed by two-sample Wilcoxon rank-sum tests with continuity correction. Correction of multiple comparisons was done with the Bonferroni method. Significance was assumed in cases with corrected *p*-values < 0.05.

### 2.7. Isolation of Microsomes from Murine Liver Tissue and Enzymatic Digestion of BaP

C57BL/6 mice were i.p. injected with 200 mg/kg body weight BaP. After 20 h, mice were sacrificed and the liver was isolated, transferred in ice-cold microsome buffer (25 mM Sucrose, 1 mM EDTA, 1 mM NADPH; 1:5, *w/v*) and homogenized for 3 min at 4 °C at maximum power (Ultra-Turrax^®^, IKA^®^-Werke). Unsolved compartments were spun down (12,000× *g*, 20 min, 4 °C) and the supernatant was transferred into a fresh vial; 100 µL BaP stock solution (10 mg/mL) was added to 5 mL supernatant (final conc. [0.2 mg/mL) and incubated at 37 °C on a conventional shaking incubator at 220 rpm (Innova40, New Brunswick Scientific, Edison, NJ, USA). After 2, 4, 6 and 8 h, 1 mL sample per reaction was harvested, aliquoted into two Eppendorf vials and shock-frozen in liquid nitrogen. Microsome fraction was released by 3 freeze/thaw cycles and subsequent sonication 5 times for 3 s on ice at half-maximum power (Sonicator 3000, Misonix, Farmingdale, NY, USA). Then, the sample was added to 1.5 parts (*v/v*) ethyl acetate, mixed and spun down (9300× *g*, 1 min, 4 °C). The organic phase was harvested using a Hamilton microliter syringe and transferred into a GC glass vial. After removing ethyl acetate using a vacuum centrifuge, BaP metabolites were dissolved in DMSO at a final concentration of 5 µg/mL.

### 2.8. Characterization of Bacterial Growth Behavior In Vitro

S.E. (0.2% from an overnight-pre-culture) were inoculated into a 24-well cell culture plate containing 1 mL LB medium per well and different concentrations of BaP or BaP metabolites that were obtained from 2-, 4-, 6- or 8-h enzymatic digest of BaP using liver microsomes. The solvent dimethyl sulfoxide (DMSO, 0.04% or 0.5%, respectively) and the antibiotic gentamycin (100 µg/mL) diluted in medium or PBS were used as negative or positive controls, respectively. Bacteria were cultured at 37 °C until the end of the log-phase while shaking (220 rpm; Shaking incubator Innova40, New Brunswick Scientific). Optical density at a wavelength of 600 nm (OD_600_) was measured every 30 min using a conventional plate reader (Safire2™, Tecan Group, Männedorf, Switzerland).

Furthermore, potential growth inhibition of salmonellae by BaP or BaP metabolites on agar was tested by plating 1 × 10^7^, 5 × 10^6^ and 1 × 10^6^ CFU log-phase S.E. (diluted in LB medium) onto LB or XLD agar in sterile disposable 10 cm diameter petri dishes. Sterilized filter paper discs (Ø 5 mm) were distributed on the agar surface and 20 µL of BaP or BaP metabolites in different concentrations dissolved in DMSO (0.04% or 0.5%, respectively) were transferred onto the filter discs. DMSO and gentamycin (100 µg/mL) diluted in medium or PBS were used as a negative or positive control, respectively. Dishes were incubated at 37 °C and analyzed for growth inhibition after 24 and 48 h.

### 2.9. Peritoneal Lavage

Mice were euthanized by CO_2_ as described above, and peritoneal exudate cells were harvested by injection of 3 mL cold phosphate-buffered saline (PBS, 0.15 M NaCl, 50 mM EDTA). After 1 min of incubation and shaking the peritoneum, the cells were aspirated. Harvested peritoneal cavity cells (PCC) were centrifuged, washed twice in cold PBS, counted and prepared for the following experiments (i.e., flow cytometry, qRT-PCR).

### 2.10. Incubation of Peritoneal Cavity Cells and Splenocytes with hk S.E.

After isolation of PCCs and spleen cells from *Ahr*^+/+^ and *Ahr*^−/−^ mice, 1 × 10^6^ cells/mL were seeded in complete RPMI 1640 culture medium (supplements see above). In addition, the cells were stimulated with heat-killed *S. enterica* Serovar Enteritidis (hk S.E.) for 20 h and subsequently analyzed by flow cytometry for expression of selected surface molecules and secretion of cytokines.

### 2.11. Immunophenotyping of Peritoneal Cavity Cells and Splenocytes

Prior to immunophenotyping, cells were incubated with rat anti-mouse TruStain fcX™ (BioLegend, London, UK) for 20 min in order to block non-specific Fc-receptor-mediated staining. For functional phenotyping, cells were stained with the following combinations of fluorochrome-labeled antibodies: anti-CD11c (clone N418), anti-I-A^b^ (MHC class II, clone AF-120.1), anti-F4/80 (clone BM8), anti-CD64 (FcγRI, clone X54-5/7.1), anti-CD80 (clone 16-10A1), anti-CD86 (clone GL-1) (all from BioLegend), anti-CD11b (clone M1/70), anti-Gr-1/Ly-6c (clone RB6-8C5), anti-CD14 (clone Sa2-8) (both from eBioscience, Frankfurt, Germany) for 30 min. Isotype-matched immunoglobulins (BioLegend or eBioscience) were used to assess non-specific staining. All incubation steps were carried out at 4 °C and protected from light. Fluorescence intensity was measured by flow cytometry (Cytomics FC 500, Beckmann Coulter, Krefeld, Germany). Analysis of flow cytometry data was performed using FlowJo™ 7.6.5 software program (Tree Star, Ashland, OR, USA).

### 2.12. Cytokine Determination in Peritoneal Lavage Fluid, Spleen Cell Culture Supernatants or Sera

Cell-free supernatants of peritoneal lavage fluid or ex vivo S.E.-antigen-stimulated splenocytes, as well as mouse sera, were collected and analyzed for relevant cytokines by ELISA. For determination of IL-1β, IL-6, IL-10, IL-23, IFN-γ and TNF-α ELISA reagent sets and protocols were purchased from R&D Systems (Wiesbaden, Germany). Precoated plates were incubated with samples and detection antibodies according to the manufacturer’s instructions. Finally, the optical density (OD450) was quantified using a conventional microplate reader (Tecan Safire2; Tecan Group, Männedorf, Switzerland). In some experiments, IL-1β, TNF-α, IL-4, IL-6, IL-10, IL-12, IL-17A, IFN-γ, MIP-1α and GM-CSF were determined by Cytometric Bead Array (CBA, Becton Dickinson, Heidelberg, Germany) according to manufacturer’s instructions. Samples were analyzed by flow cytometry (FACSArray™, Becton Dickinson).

### 2.13. Serum Immunoglobulin Analysis

*Salmonella*-antigen specific IgG1 and IgG2c serum antibodies were captured by plate-bound S.E.-antigen (coated overnight at 4 °C in 50 mM carbonate buffer, pH 9.6, on PolySorp™ plates, Nunc, Wiesbaden, Germany) for 1.5 h at room temperature (RT), washed 3 times with PBS (0.15 M NaCl, 0.05% Tween^®^ 20, (PBS-T)), blocked with blocking buffer (0.3 M NaCl, 0.5% Casein, 0.05% Tween^®^ 20) and detected by the respective subisotype-specific peroxidase-conjugated polyclonal detection antibody (1:5000; purchased from Southern Biotechnology, Birmingham, AL, USA) for 1 h at RT. Signal was generated by incubation with TMB (3, 3′, 5, 5′-tetramethylbenzidine) ELISA peroxidase substrate buffer (Moss, Pasadena, CA, USA). Absorption was quantified as OD_450_ using a conventional microplate reader (Safire2™, Tecan Group). The preparation of the S.E. antigen was described elsewhere [28,29]. These in-house ELISAs were validated using sera from S.E.-infected and non-infected control SPF mice. The lower limit of detection for both ELISAs was titer 50. The cut-off values were determined as the mean OD of the negative SPF sera +3 × SD. The developed tests revealed very low background titer in SPF animals and did not cross-react with antibodies against other enterobacteria such as *E. coli*.

### 2.14. Protein Separation and Western Blot

Frozen spleen samples from S.E.-infected or healthy, non-infected C57BL/6 mice were thawed on ice and homogenized in 200 µL RIPA lysis buffer (1 mM PMSF, 10 mM DTT, 40 µL protease inhibitor (Roche Diagnostics, Penzberg, Germany), leupeptin hemisulfat, pepstatin A, aprotinin) for 10 min at 4 °C (Ultra-Turrax^®^, IKA^®^-Werke). Unsolved compartments were spun down (12,000× *g*, 20 min, 4 °C) and supernatant was transferred into a fresh vial. After determination of total protein concentration by the Bradford method, lysates were separated by SDS-PAGE (10%) and transferred onto a PVDF membrane using a PerfectBlue Semi-Dry blotting device (Peqlab Biotechnologie, Erlangen, Germany) at 80 mA for 2 h. Membranes were blocked in blocking buffer (Tris-buffered saline, 0.05% Tween20 (TBS-T), 4% *w*/*v* milk powder) overnight at 4 °C followed by incubation with primary antibody (Rat anti-mouse AhR, clone RPT1, Abcam, 1:1000 or rat anti-mouse GAPDH (# G9545, Sigma Aldrich) 1:5000) in blocking buffer for 1.5 h at RT. Subsequently, membranes were washed 4 times in blocking buffer and incubated with HRP-conjugated secondary antibody (F(ab’)_2_ fragment goat anti-rat IgG (H + L); # 112-036-062, Jackson ImmunoResearch, Ely, UK) 1:10,000 in blocking buffer for 1 h at RT. Finally, membranes were washed 3 times in TBS-T and once in TBS.

The chemiluminescence signal was generated using the ECL Western blotting detection reagents and analysis system (GE Healthcare, Freiburg, Germany) according to the manufacturer’s instructions and membranes were exposed to X-ray film (GE Healthcare) and analyzed using a Gel iX Imager (INTAS Science Imaging Instruments, Göttingen, Germany). For semi-quantitative assessment, membranes were stripped in 20 mL stripping solution (65 mM Tris/HCL, pH 6.7, 2% SDS, 140 µL β-mercaptoethanol) for 20 min at 50 °C and subsequently washed 4 × 10 min in TBS-T. Then, the stripped membrane was stained for the housekeeping protein GAPDH.

### 2.15. Determination of Nitric Oxide (NO) Synthesis

The concentration of nitrite and nitrate as a stable final product of NO synthesis was determined in supernatants of peritoneal lavage fluid, as described by Green et al. [31], using the Griess reaction. A standard curve was prepared from serial dilutions of a sodium nitrate stock solution. Fifty microliters of cell-free supernatant were mixed with 100 µL of Griess reagent (0.1% *N*-(1-naphthyl)-ethylenediamine-dihydrochloride in absolute ethanol, 1% sulfanilamide in 5% phosphoric acid) and incubated for 10 min at RT. Concentration of nitrite/nitrate was quantified as OD_550_ using a conventional microplate reader (Safire2™, Tecan Group).

### 2.16. Determination of Phagocytic Activity

PCCs were tested for their ability to ingest fluorescein-isothiocyanate (FITC)-labeled hk S.E that is used as standard antigen in our model. Hk S.E. were incubated with FITC (0.1 mg/mL) in carbonate buffer (0.1 M) for 1 h at room temperature, washed three times with PBS and stored at −20 °C until use. After isolation, PCCs were incubated for 1 h with FITC-labeled hk S.E. at 37 °C. Negative controls were performed simultaneously by incubating cells on ice instead of 37 °C. The uptake of FITC-labeled hk S.E. was assessed by flow cytometry using a conventional flow cytometer (FC 500, Beckmann Coulter) and analyzed using the software program FlowJo™ 7.6.5 (Tree Star).

### 2.17. Quantification of mRNA Expression via Real-Time RT-PCR

Total RNA was isolated from peritoneal or spleen cells by means of TriReagent^®^ according to the manufacturer’s instructions (Sigma Aldrich). After quantifying by absorbance at 260 nm/280 nm, DNA contamination was degraded by DNaseI (Fermentas, St. Leon Rot, France). The single-strand cDNA synthesis was performed using a Transcriptor First Strand cDNA Synthesis kit (Roche Diagnostics). Optimal primer design and probe selection for mRNA detection of selected genes from the Universal ProbeLibrary^®^ (UPL; Roche Diagnostics) was performed using the web-based software tool ProbeFinder (Roche Diagnostics) (Table 1). Designed primers were purchased from TibMolBiol (Berlin, Germany) and real-time RT-PCR was performed using the LightCycler^®^ 480 instrument (Roche Diagnostics). PCR assays were prepared using LightCycler^®^ Probes Master kit (Roche) and the appropriate UPL probe with optimal primers. As housekeeping genes, aminolevulinate synthase 1 (*Alas1*) and hypoxanthine-guanine phosphoribosyl transferase (*Hprt*) were used to normalize the gene expression results. Relative quantification was performed by means of the LightCycler^®^ 480 software v.1.5 (Roche).

### 2.18. Statistical Analysis

Significant differences between datasets were estimated by Kruskal–Wallis one-way ANOVA using SigmaPlot™ 14 software (Systat, Erkrath, Germany). In the case that the normality test (Shapiro-Wilk method) and the equal variance test (Brown–Forsythe method) passed, the Holm–Sidak test for all pairwise multiple comparison procedures was applied post-hoc, and values of *p* < 0.05 were considered significantly different (* *p* ≤ 0.05; ** *p* ≤ 0.005; *** *p* ≤ 0.001).

Differences between datasets were tested for statistical significance by Kruskal–Wallis one-way ANOVA or ANOVA on ranks (if normality and/or equal variance tests failed) followed by the Holm–Sidak or Dunn’s/Tukey’s post-hoc analysis, respectively (* *p* ≤ 0.05, ** *p* ≤ 0.01, *** *p* ≤ 0.001). In specified cases, the means or medians of two samples (e.g., comparison *Ahr* wild-type vs. *Ahr*^−/−^ mice) were compared by Student’s or Welch’s *t*-test (dependent on the equal variance) if the normality and equal variance test passed or by the Mann–Whitney rank-sum test (*u*-test) if the normality and/or equal variance test failed.

## 3. Results

### 3.1. BaP Improves the Survival of Mice Infected with S. enteritidis

The intraperitoneal infection of mice with *S.* Enteritidis caused septic peritonitis with high morbidity and mortality due to secondary multiorgan dysfunction. The application of both BaP doses (i.e., 20 ng/kg and 2 µg/kg bw) and the application of the non-toxic AhR ligand FICZ at a dose of 1.25 µg/kg bw improved the survival of infected mice. The mortality of mice treated with the vehicle control corn oil increased continuously in the period between 15 to 40 days post infection (p.i.), whereas the mortality rate of BaP-exposed mice increased significantly lower. In detail, 90 days p.i., only 5% of mice with vehicle control survived the infection, while treatment with the higher dose of BaP improved survival up to 60% and the lower dose up to 80% (Figure 1).

### 3.2. Impaired Bactericidal Capacity and Persistent Infection in BaP-Exposed Mice

Intraperitoneal infection of C57BL/6 mice with S.E. resulted in rapid dissemination of the bacteria into the visceral organs, predominantly spleen and liver. The methodology chosen for this study allows estimation of the total salmonellae burden per organ, and moreover, the number of phagocytes that had engulfed salmonellae. This parameter was determined in the peritoneal cavity (compartment of local infection, peritonitis) and in the spleen (compartment of systemic infection). BaP-exposure of mice had no influence on bacterial loads in both spleen (Figure 2a) and liver (Figure 2b) up to day 30 p.i. However, at day 90 p.i. the salmonellae burden was significantly higher in both spleen and liver from mice treated with 0.02 µg/kg bw BaP and in the liver from mice treated with 2 µg/kg bw BaP compared to the vehicle control, suggesting that long-term BaP exposure led to persistent infection in that model.

Investigation of the number of S.E.-positive splenocytes, most likely representing phagocytes, revealed a statistically higher number of S.E.-positive cells in the early infection phase (i.e., days 3 and 10 p.i.) under exposure to 0.02 µg/kg bw BaP. However, this difference could no longer be observed on day 20 p.i. After day 20 p.i., the numbers of infected splenocytes decreased in all three groups, while the strongest decrease was found in mice exposed to 2 µg/kg bw BaP, showing significantly lower amounts on day 30 p.i. than found in the vehicle control (Figure 3).

Thus, there were two important results from these experiments, (i) higher survival of salmonellae in splenic phagocytes from BaP-exposed mice in the early infection phase and (ii) persistence of salmonellae in spleen and liver until day 90 p.i. seems to be due to extracellular rather than intracellular survival of the bacteria.

### 3.3. Bioavailability of BaP and Its Most Prominent Metabolites in the Liver

The influence of BaP on the immune response to *S. enterica* was studied at two different doses of BaP, i.e., 20 ng/kg and 2 µg/kg bw. When administered at a dose of 20 ng/kg bw, BaP was completely metabolized within 3 days of exposure. Using flow injection analysis MS/MS, 4 out of 8 investigated BaP metabolites could be detected in liver homogenates derived from BaP-exposed C57BL/6 mice, while BaP was no longer detectable. The most prominent metabolite was 8-Hydroxy-BaP (Figure 4a). After 30 days of exposure with the same BaP dose, low concentrations of non-metabolized BaP could be detected in mouse livers. However, in this case, 8-Hydroxy-BaP was found to be the most abundant metabolite (Figure 4b). When mice had been exposed with 2 µg/kg bw BaP for 3 days, 27.7 nmol non-metabolized BaP per g liver tissue were detected (MW ± SEM, n = 3; not shown).

### 3.4. Neither BaP nor BaP Metabolites Affect Survival and Growth Behavior of S.E.

Since administration of BaP alone, i.e., without infection with S.E., had only little effect on the immune parameters determined in this study, direct effects of BaP onto the salmonellae could not be excluded in principle and had to be considered for the explanation of the observed phenotype. Therefore, we have studied those potential bactericidal, bacteriostatic or possibly even growth-promoting effects of BaP or BaP metabolites on the *S. enterica* strain used in our model in vitro using several assays (i.e., agar diffusion assays in LB and XLD agar, shown for LB medium in Figure 5a–d, log-culture assay in LB medium, Figure 5e,f, and growth behavior on selective XLD agar, Figure 5g). BaP- or BaP-metabolite mediated effects on the viability or the growth behavior of salmonellae were not detectable in any of these assays. Neither the number nor the shape of S.E. colonies on selective XLD agar were affected by BaP or BaP metabolites in comparison to the solvent control DMSO. Hence, this result implicated with the utmost probability that BaP-induced effects onto certain immune cells and functions during the ongoing immune response against *S. enterica* rather than direct effects onto the bacteria were responsible for the central findings of this study.

### 3.5. CD14, MHC Class II and FcγR1 Induction through BaP in Peritoneal Innate Immune Cells

On day 3 post-S.E. infection, staining of cells isolated by peritoneal lavage with common fluorescence-labeled phenotypic markers and flow cytometric analysis revealed two dominant innate immune cell populations: (1) CD11b^+^/Gr1(Ly-6G/Ly-6C)^+^ and (2) CD11b^high^/F4/80^+^. Although these cell populations were not immunophenotyped in more detail, we considered population one to be mainly granulocytes and population two to be mainly macrophages and monocytes as confirmed by panoptic staining according to Pappenheim (Figure 6a). Lower abundant cell populations in the peritoneal lavage fluid were shown to be CD11b^high^/CD11c^+^ (considered to be mainly dendritic cells), CD3^−^/NK1.1^+^/NKp46^+^ (considered to be NK cells) (Table 2) and CD3^+^/γδTCR^+^ (considered to be γδT cells; not shown). To characterize the functional role of these cell populations and its potential modulation by BaP in the early phase of S.E. infection, we studied several functionally relevant surface markers (i.e., I-A^b^, CD14, CD64, CD80, CD86) by flow cytometry. The results reveal several alterations on peritoneal innate immune cells through BaP exposure. CD11b^+^/Gr1^+^ cells expressed significantly higher amounts of CD14 under exposure with 2 µg/kg bw BaP, which was apparent on day 1 p.i. in the spleen (Table 3) and on day 3 p.i. in the peritoneal cavity (Figure 6b). Remarkably, CD64, a molecule that is not constitutively expressed on neutrophils, could be detected on CD11b^+^/Gr1^+^ granulocytes from the spleen at day 1 p.i. (Table 3) and from the peritoneal cavity (day 3 p.i., Figure 6b) in mice exposed to 2 µg/kg bw BaP. At day 3 p.i., the surface expression rates of the murine MHC class II alloantigen I-A^b^ and the high-affinity IgG receptor FcγRI (CD64) were significantly elevated through 2 µg/kg bw BaP on CD11b^high^/F4/80^+^ cells, while the increase of both activation markers through 20 ng/kg bw BaP was not statistically significant. However, on day 1 p.i., the I-A^b^ expression rate was found to be induced by 20 ng/kg bw in the spleen (Figure 6c, Table 3). Of note, at day 1 p.i., also the proportion of CD64-expressing splenocytes was significantly elevated through 2 µg/kg bw BaP up to **71.6**
**± 5.1%** (p ≤ 0.05) vs. 62.8 ± 3.7% (vehicle control) and 65.5 ± 8.1% (0.02 µg/kg bw BaP). Furthermore, the I-A^b^ expression was also significantly upregulated on CD11b^high^/CD11c^+^ cells in BaP-exposed mice at day 3 p.i. through both applied BaP doses (Figure 6d, Table 3).

### 3.6. BaP Alters Functional Properties of Peritoneal Cavity Cells

After i.p. infection of C57BL/6 mice, the resident and recruited innate immune cells of the peritoneal cavity represent the first line of defense against invading salmonellae. In order to further characterize the functional properties of innate immune cells at the site of infection, we analyzed the mRNA expression of relevant pro- and anti-inflammatory cytokines (i.e., *Il1b*, *Tnfa*, *Il10*), as well as critical functional features of phagocytes (i.e., phagocytosis rate, nitric oxide (NO) production). The mRNA results show that the transcription of *Il1b* was not significantly altered at day 1 p.i., but significantly increased at day 3 p.i. through exposure with the higher BaP dose (2 µg/kg bw; Figure 7a). The transcription rate of *Il1b* was also slightly increased by the lower BaP dose (20 ng/kg bw) and the endogenous AhR ligand FICZ as well. However, these results were not statistically significant. Neither the results achieved for *Tnfa* nor for *Il10* revealed statistically significant differences (Figure 7b,c). The mRNA data were found to be only in part consistent with the protein data achieved for days 1 and 3 p.i. In the case of IL-1β, the protein concentration in the peritoneal lavage fluid was significantly decreased by the higher BaP dose and FICZ at day 1 p.i., but significantly increased at day 3 p.i. by the higher BaP dose (Figure 7d). TNF-α was decreased by BaP but interestingly not by FICZ at day 1 p.i. On day 3 p.i., the TNF-α concentration was elevated in all groups compared to the levels detected on day 1 p.i. without significant differences between the four groups (Figure 7e). In contrast to the proinflammatory cytokines IL-1β, TNF-α (both at day 1 p.i.) and IL-6 (at day 3 p.i., not shown), all of which were significantly decreased in the peritoneal cavity at the protein level, the anti-inflammatory cytokine IL-10 was significantly increased by 2 µg/kg BaP as well as FICZ at day 1 p.i. (Figure 7f).

Furthermore, we assessed the effect of BaP treatment on the uptake of hk S.E. by isolated peritoneal cavity cells (PCCs). BaP treatment increased the amount of internalized FITC-labeled hk S.E. by peritoneal phagocytes, as demonstrated exemplarily by flow cytometry and fluorescence microscopy (Figure 8a, Table 4). This result was statistically significant in terms of the lower BaP dose (20 ng/kg bw) at day 1 p.i. (Figure 8b, left graph) or in terms of the higher BaP dose (2 µg/kg bw) and FICZ at day 3 p.i. (Figure 8b, right graph).

Upon uptake of intracellular residing bacteria and activation by proinflammatory cytokines (i.e., TNF-α, IL-1β, IFN-γ), macrophages and granulocytes can produce NO through inducible nitric oxide synthase (iNOS). Since NO is highly cytotoxic to intracellular pathogens and thus may be a critical parameter that might be affected by BaP, we investigated NO production of PCCs in the supernatant of peritoneal lavage fluids. On day 3 p.i., PCCs produced higher amounts of NO when exposed to both BaP concentrations and FICZ in comparison to the vehicle control. However, the difference between the lower BaP dose (20 ng/kg bw) and the vehicle control was not statistically significant (Figure 8c). Interestingly, if isolated PCCs were stimulated ex vivo with 2 × 10^7^ CFU hk S.E., cells from the vehicle control produced significantly higher amounts of NO over a period of 46 h (Figure 8d). The results from Figure 8c directly indicate the NO concentration in the peritoneal cavity. The nitrite amounts measured ex vivo represent the capacity of NO production by peritoneal phagocytes induced by phagocytosis of living salmonellae, mainly as a response to macrophage activation by IFN-γ and TNF-α. However, Figure 8d represents data from peritoneal cells from healthy, non-infected animals that were stimulated ex vivo with S.E. antigen (i.e., heat-killed S.E., offers a broad range of pathogen-associated molecular patterns). Although co-stimulation with BaP and hk S.E. was capable of inducing NO over time, this effect was superposed by the corn oil result. Although administrating polycyclic hydrocarbons, such as BaP, dissolved in corn oil is the method of choice, this procedure has some disadvantages. Corn oil, like many other oils, alone induces a sterile inflammation in the peritoneal cavity. Thus, some immunologic effects induced by corn oil could hardly be discriminated from the BaP effect. The NO results from the peritoneal cavity fluid are easier to interpret since the detected in vivo NO production induced by S.E. infection clearly demonstrates an AhR-ligand-dependent effect apart from the effect mediated by corn oil. Considering the dilution effect associated with the peritoneal lavage, the real nitrite concentration in the peritoneal cavity (i.e., the site of infection) is proportionally higher in all groups.

### 3.7. S.E.-Specific Antibody Response Is Increased through BaP Exposure

Although cellular immunity is of crucial importance for the control of salmonellae and also salmonellae-specific antibodies, predominant antibodies of the IgG subclass 2c, but also those of subclass IgG1, are essential for sterile elimination of salmonellae in our model. Therefore, the S.E.-specific IgG2c and IgG1 titers were analyzed throughout the course of infection under BaP exposure (Figure 9). As expected, the S.E.-specific IgG1 serum titer was almost 2 log stages below the IgG2c titer. While the IgG1 titers were comparably low in most animals of all three groups at day 20 p.i.; at day 30 p.i., the S.E.-specific IgG1 titers were found to be elevated in the BaP-exposed groups. However, this result was statistically significant only for the higher BaP dose (2 µg/kg bw). After 90 days p.i., the IgG1 titer in mice treated with 20 ng/kg bw BaP was significantly higher than in the vehicle control group (Figure 9a). The IgG2c serum titer was elevated in BaP-exposed C57BL/6 mice compared to the vehicle control as early as day 20 p.i., which was statistically significant for the higher BaP dose (2 µg/kg bw). Interestingly, there was no difference between the three groups on day 30 p.i. (at the end of the acute infection phase), but in the late infection phase (day 90 p.i.), the S.E.-specific IgG2c titers were found to be further increased. Both BaP doses caused significantly higher IgG2c titers compared to the vehicle control (Figure 9b).

### 3.8. BaP Alters Cytokine Concentration in Sera

In order to assess the impact of BaP exposure on the systemic immune response to S.E., we have analyzed the serum levels of relevant cytokines in S.E.-infected BaP-exposed C57BL/6 mice in the early, ongoing (day 3 p.i.) and established phases of S.E. infection (day 10 p.i.). Similar to the protein data from the peritoneal fluid, also in the sera from infected mice, proinflammatory cytokines were found to be reduced by BaP at day 10 p.i. (i.e., IL-6, IL-12), while TNF-α and IFN-γ were not significantly affected. In contrast, serum concentrations of the anti-inflammatory cytokines Il-4 and IL-10 were significantly elevated through the lower BaP dose (20 ng/kg bw) at days 3 or 10 p.i., respectively (Figure 10).

### 3.9. BaP-Induced Immune Modulation Is Mainly AhR-Dependent

To test whether the observed effects of BaP exposure in S.E.-infected C57BL/6 mice were AhR-dependent, we compared selected critical parameters in Ahr wild-type (Ahr^+/+^) and Ahr knock-out (Ahr^−/−^) mice, both on the C57BL/6 background (Figure 11). In this study, immune cells from the peritoneal cavity and spleen were isolated after 20 h exposure with 2 µg/kg bw BaP. Ex vivo, the cells were stimulated with 2 × 10^7^ CFU hk S.E. for another 20 h. Subsequently, the concentration of the anti-inflammatory cytokine IL-10 and the proinflammatory cytokines TNF-α or IL-23 in culture supernatants from peritoneal cavity or spleen cells, respectively, was determined by ELISA. Finally, the cells were harvested for analysis of the MHC class II (I-A^b^) surface expression by flow cytometry.

In both peritoneal and spleen cells from wild-type mice, the S.E.-antigen-induced production of IL-10 was found to be significantly enhanced through BaP exposure in vivo prior to antigen stimulation ex vivo. In contrast, the IL-10 concentration in supernatants of peritoneal and spleen cells from Ahr^−/−^ mice was found at the same level as the vehicle control (Figure 11a,b). Hence, the observed enhancement of S.E.-antigen-induced IL-10 production of immune cells at the site of infection, as well as systemically in the spleen, through BaP was AhR-dependent. A very similar result was obtained in terms of the MHC class II (I-A^b^) expression rate on spleen cells, which was significantly enhanced by BaP pre-exposure in an AhR-dependent manner (Figure 11f). Although the I-A^b^ expression rate on peritoneal cavity cells from wild-type mice was also found to be significantly enhanced after BaP exposure, this was obviously not AhR-dependent because the effect could not be abolished on cells from Ahr^−/−^ mice. In the spleen, IL-23 was analyzed as a proinflammatory cytokine that is crucially involved in protective immunity to *Salmonella enterica*. While the S.E.-antigen-induced IL-23 production of splenocytes derived from wild-type mice ex vivo was significantly impaired after pre-exposure to BaP in vivo, splenocytes derived from Ahr^−/−^ mice secreted nearly the same IL-23 amount as splenocytes from wild-type mice but interestingly on the same low level found in wild-type mice following BaP pre-exposure (Figure 11d). This result let us assume that S.E.-induced IL-23 production in splenocytes is dependent on AhR activation. More complex appears the explanation for the TNF-α result. The detected amount of TNF-α in culture supernatants of S.E.-antigen stimulated peritoneal cells was found to be significantly decreased in the vehicle control compared to cells from BaP-exposed wild-type mice. However, this phenomenon was found to be completely restored in S.E.-stimulated peritoneal cells from Ahr^−/−^ mice (Figure 11e). This means that the observed TNF-α suppression in peritoneal inflammatory cells from wild-type mice was AhR-dependent but independent on BaP exposure.

### 3.10. BaP Activates AhR and Induced Both Canonical and Non-Canonical AhR Signaling Pathways

To verify the activation of AhR by BaP and the subsequent induction of the canonical or the non-canonical AhR signaling pathway in healthy (non-infected) and S.E.-infected C57BL/6 mice, we studied the mRNA expression of Cyp1a1, Ahr, Ahrr and Il22 or Nfkb1, Rela, Tgfb1 and Il10, respectively, by quantitative RT-PCR (Figure 12a,c–f and Figure 13c,d). In S.E.-infected mice, Ahr transcription was significantly induced through 2 µg/kg bw BaP and 1.25 µg/kg bw FICZ at day 1 p.i. (Figure 12a). Although Ahrr mRNA expression was induced dose-dependently by BaP at day 3 p.i., the differences to the result found in the vehicle control were not statistically significant. Interestingly, the difference was significant when compared to the FICZ result since FICZ did not induce Ahrr (Figure 12c). Cyp1a1 (Figure 12d) and Il22 (Figure 13c) were significantly upregulated on the transcriptional level after administration of 2 µg/kg bw BaP in S.E. infected C57BL/6 mice at day 3 p.i. Together, these data suggest the AhR ligation by BaP has activated the canonical AhR signaling pathway. However, Nfkb1 (Figure 12e), Rela (Figure 12f), Tgfb, and Il10 (Figure 13d) were also significantly upregulated in response to BaP exposure in S.E. infected C57BL/6 mice at day 3 p.i., suggesting the activation of the non-canonical AhR signaling pathway.

Moreover, the AhR protein expression in spleen cells from healthy (i.e., non-infected) and S.E.-infected C57BL/6 mice was semi-quantitatively analyzed by Western blot. The result supported the mRNA data. Moreover, on the protein level, AhR was significantly induced by BaP in S.E.-infected mice, while only a very weak AhR signal was detected in non-infected mice (Figure 12b).

### 3.11. mRNA Expression Patterns of Spleen Cells from BaP Exposed C57BL/6 Mice

As a first approach to correlate the unexpected clinical phenotype of preventing the fatal outcome of septic peritonitis through BaP exposure with a special systemic immune phenotype in these animals, we have performed a series of quantitative RT-PCR analyses using purified and quality-approved total RNA samples from spleen cells derived from BaP-exposed vs. corn-oil treated mice (vehicle control). These experiments were performed on day 3 p.i. (Figure 13) and day 10 p.i. (Figure 14). Hence, the obtained results may characterize both the early innate and the ongoing adaptive immune response to S.E. The analyzed genes were classified into five different expression panels: a Th1-like panel, represented by *Il12a*, *Il12b*, *Ifng*, and *Tbx21* (Figure 13a and Figure 14a), a Th2-like panel represented by *Il4*, *Il13*, and *Gata3* (Figure 13b and Figure 14b), an ILC3/Th17/Th22-like panel represented by *Il17a*, *Rorc*, *Il22*, and *Ahr* (Figure 13c and Figure 14c), a Treg-like panel represented by *Il6*, *Tgfb1*, *Il10*, and *Fop3* (Figure 13d and Figure 14d) and an inflammatory panel represented by *Il1b*, *Tnf*, and *Il18* (Figure 13e and Figure 14e).

In the early phase of S.E. infection (i.e., day 3 p.i.), we detected significant induction of genes from the Th1-like panel (*Il12b*, *Tbx21*), from the Th2-like panel (*Gata3*), from the ILC3/Th17/Th22-like panel (*Il22*, *Ahr*), from the Treg-like panel (*Il6*, *Il10*) and from the inflammatory panel (*Il1b*, *Il18*) under BaP exposure with one or both doses compared to the vehicle control. Furthermore, *Gata3* was found to be downregulated in S.E.-infected compared to non-infected mice, which was completely restored or even upregulated by BaP exposure (Figure 13b). In S.E.-infected mice, the proinflammatory cytokines *Il1b* and *Il18* were upregulated compared to the vehicle control under exposure with 2 µg/kg bw BaP or with both BaP doses, respectively. In contrast, *Tnf* was significantly upregulated after treatment with 2 µg/kg bw BaP in non-infected mice, while no differences were detected in S.E.-infected animals (Figure 13e).

Under BaP exposure, most of the genes that were found to be upregulated at day 3 p.i. were no longer upregulated at day 10 p.i., i.e., in the ongoing adaptive immune response phase. In contrast, several genes were found to be downregulated in S.E.-infected mice with or without BaP exposure. The only exceptions were *Il12a*, *Ifng*, *Il17a*, and *Il22*. *Il12a* was significantly induced only through the higher BaP dose with or without S.E. infection, when compared to the respective vehicle control, while the *Ifng* mRNA expression level was significantly higher in S.E.-infected mice compared to non-infected mice after exposure with both BaP doses (Figure 14a). Only a weak but significant transcriptional induction of *Il17a* and *Il22* was found in S.E.-infected compared to non-infected mice only under exposure with the lower BaP dose (Figure 14c).

Although a clear immunological phenotype could not be drawn from these results, the data revealed a significant influence of BaP on the expression of several cytokine genes. Among them were proinflammatory, anti-inflammatory and regulatory cytokines. Of note, the BaP-dependent expression of these cytokine genes differed between S.E.-infected and non-infected mice, suggesting that BaP exerted different effects in healthy and infected animals. In future in-depth studies, it should be elucidated whether BaP may crucially influence the activity of important innate immune cells (e.g., ILCs, NK cells) and/or the differentiation of Th cells. In particular, it has to be elucidated whether the higher induction of IL-12 and IFN-γ under the influence of BaP in the early infection phase might be a crucial condition for the control of a bacterial infection. Moreover, the cellular sources of these two cytokines should be identified to explain the observed clinical phenotype in our S.E. infection model. Furthermore, the importance of the regulatory cytokines IL-6, IL-10 and IL-22 for the control of the strong cellular immune response against S.E. has to be studied in more detail.

## 4. Discussion

This study was designed to determine the influence of BaP on the outcome of systemic *Salmonella enterica* infection. In humans, foodborne infection with the *S. enterica* serovars Typhi and Paratyphi tends to cause a serious and life-threatening infection called typhoid or paratyphoid fever, respectively [32,33]. Invasive nontyphoidal *Salmonella* serovars, e.g., *S. enterica* serovars Typhimurium and Enteritidis, are a leading cause of lethal sepsis and severe relapsing infections in young children and immunocompromised individuals [34]. Previously, our group has developed and comprehensively characterized a mouse model of systemic *S. enterica* infection induced by intraperitoneal infection of BALB/c or C57BL/6 mice with a live attenuated strain of *S. enterica*, serovar Enteritidis (S.E.) [28], that is used as a vaccine strain in chicken and swine [26,27]. Therefore, this model has previously been utilized for efficacy and safety testing of *Salmonella* vaccines [29]. Moreover, in terms of pathogenesis research and therapy validation, it can also be used as a disease model of septic peritonitis. Immunotoxic adverse effects exerted by many xenobiotics can hardly be recognized in healthy animals due to the fact that acute toxic effects become apparent only in relatively high doses [35]. Thus, immunotoxic effects of BaP reported in the literature were observed at higher doses up to 50 mg/kg body weight [35,36]. Moreover, the normal immune status in healthy individuals is commonly not very fragile; hence, strong adverse effects are needed to alter immune functions at baseline. However, weak or chronic immunotoxic effects of xenobiotics or drugs may have a stronger impact under certain disease conditions. Hence, those effects can be detected with higher sensitivity in disease or vaccination/challenge in vivo models. For that reason, a previously developed mouse model of systemic *Salmonella* infection has been adapted for immunotoxicological assessment of heavy metals or other xenobiotics [37]. In the present study, this mouse model was utilized to investigate potential immunotoxic effects of the xenobiotic AhR ligand BaP in comparison to the non-toxic, endogenous AhR ligand FICZ.

Most interestingly, BaP exposure during S.E.-induced septic peritonitis in C57BL/6 mice significantly reduced the mortality of infected mice. This result is in accordance with previous observations published by Vorderstraße et al. showing that application of TCDD improves the survival of mice during experimental *Streptococcus pneumoniae* infection [19]. Furthermore, Kimura et al. have shown that the endogenous AhR ligand FICZ protects mice infected with *Listeria monocytogenes* against lethal challenges [17]. In our model, surprisingly, decreased mortality rate in BaP-exposed mice was associated with the reduced microbicidal capacity of phagocytes in the spleen during the ongoing infection, as demonstrated by higher numbers of S.E.^+^ splenocytes on days 3 and 10 p.i., and higher bacterial burden in visceral organs in the very late phase of infection, i.e., day 90 p.i., when the pathogen is almost eradicated from control mice without BaP exposure. Both observations assume that the immune response to S.E. in BaP-exposed mice is less efficient compared to vehicle-treated mice. The immune response to this S.E. strain in mice, which was extensively studied previously [28,29], is characterized by a strong type-1 immune response in both the early ongoing and the established acute phases of infection. In this model, as in *Salmonella* infection in general, the IL-12/IFN-γ axis is essential for the development of effective immunity and eradication of the bacteria [28,29,38]. The major cellular players during the early phase of infection are macrophages and DCs, as professional antigen-presenting cells and sources of IL-12, but also NK cells, ILC1, NKT cells and γδT cells, as sources of IFN-γ and TNF-α [28,38,39]. IL-12 is an essential factor for NK cell and Th1 cell maturation and differentiation, while IFN-γ and TNF-α are necessary for macrophage and DC activation [28,38]. In this phase, the type-2 immune response or the IL-17/IL-22 axis is not essential [29,39]. However, in the later stage of *S. enterica* infection, the humoral immune response and IL-4-mediated immune effector mechanism seem to be important for the eradication of salmonellae [29,35,40,41]. Obviously, BaP exposure may support the induction of the IL-12/IFN-γ axis in the ongoing S.E. infection in our model, resulting in control and clearance of salmonellae in the early phase of infection. According to this assumption, the exponential growth of bacteria is prevented or at least reduced, in turn, lowering the risk of SIRS and fatal outcomes through septic shock.

The upregulation of CD64 (FcγRI) and MHC class II on antigen-presenting cells (i.e., macrophages, DCs, neutrophils) under BaP exposure were most likely induced by IFN-γ and may be beneficial in terms of the receptor-mediated phagocytosis of IgG-coated salmonellae and the presentation of S.E. antigen to T cells, respectively, since IFN-γ was shown to induce transcriptional activation of CD64 [42]. Moreover, CD64 is also induced by IL-10, which may contribute to the elevation of phagocytosis in the same way [42]. In terms of MHC-II expression, our finding is in line with reports from other groups which found upregulated MHC class II expression on non-activated and LPS-activated bone-marrow-derived DCs following exposure to toxic or dietary AhR ligands. In this model, the effect was shown to be XRE-independent [43,44]. MHC-II expression is induced by IFN-γ via the MHC class II transactivator CIITA [45] or via the NF-κB subunit RelB [46]. It was shown that activated AhR can interact with RelB followed by transcriptional activation of genes with κB sites in BMDCs [47] and that AhR activation leads to upregulated expression of RelB [48]. Thus, the AhR-dependent upregulation of MHC class II might be explained by additional transcriptional activation through the AhR/RelB-complex.

IFN-γ also enhances intracellular killing of salmonellae through the activation of iNOS that catalyzes the production of nitric oxide, a very potent bactericidal compound [38]. Indeed, a BaP-dependent increase of S.E. phagocytosis, as well as NO production, could be shown in the present study. Similarly, Neff-LaFord et al. reported on upregulation of NO under TCDD exposure in viral bronchial infection [49]. These results are supported by findings in the BMDM in vitro model that reveal AhR-dependent NO synthesis through S.E.-antigen stimulation under BaP exposure [50,51]. Amazingly, neither BaP-dependent increased phagocytosis rate nor the synthesis of NO resulted in better elimination of the pathogens from visceral organs, suggesting suppressive effects of BaP on other defense mechanisms necessary for an efficient immune response against *S. enterica* that have to be elucidated in further studies.

To investigate the influence of BaP on cytokine production in infected mice, we investigated the concentrations of certain cytokines of the peritoneal cavity and serum. The regulatory cytokine IL-6 is a proven marker of severity during the septic course, whereby high levels correlate with increased mortality in mice and humans [52,53]. Hence, the diminished IL-6 concentration in the peritoneal cavity and in serum through BaP exposure may indicate a reduced severity of septic peritonitis in our model. Kimura et al. reported that the transcriptional activity of the *IL6* gene is inhibited by a complex consisting of ligand-activated AhR, the NF-κB subunit NF-κB1 (p50) and STAT1 in murine peritoneal macrophages [54]. This might be one explanation for BaP-dependent suppression of IL-6 in our model.

Under inflammatory conditions, the anti-inflammatory cytokine IL-10 plays a crucial role in the attenuation or limitation of immune responses. Moreover, it was shown that IL-10 is a key factor in terms of protecting mice from death caused by septic peritonitis [55,56]. In comparison to IL-6, the amounts of the anti-inflammatory cytokine IL-10 were increased in both peritoneal cavity lavage fluid and serum from BaP-exposed S.E.-infected mice. Hence, increased IL-10 secretion, most likely by peritoneal macrophages, may have prevented an overflowing immune reaction (such as SIRS in humans) and septic course. Furthermore, S.E. antigen stimulation of peritoneal cells and splenocytes ex vivo revealed a BaP-mediated enhancement of IL-10 secretion that was shown to be AhR-dependent in *AhR*-deficient mice. This result is in line with our own data previously obtained in the BMDM in vitro model [50,51], and furthermore, it is also supported by other authors who reported a higher expression of IL-10 through different AhR ligands in BMDCs after activation with LPS [43,48]. Thus, IL-10 production, most likely by macrophages, appears to be a key feature of AhR activation. Recently, Zhu et al. demonstrated that AhR promotes IL-10 expression in inflammatory macrophages through a non-canonical, non-genomic Src-STAT3 signaling pathway [57]. Moreover, these authors reported that LPS recognition by TLR4, and possibly also other pattern recognition receptor ligands, promotes AhR expression in macrophages. Hence, this may explain the more extensive effects of BaP in S.E.-infected vs. healthy mice.

Similar to IL-10, IL-22, another member of the IL-10 family of cytokines, was found to be significantly induced in S.E.-infected BaP-exposed mice. Gene expression of *Il22* is induced through the canonical AhR signaling pathway [58]. The most important function of this cytokine is to control the homeostasis of microbiota on epithelial barriers through the induction of antimicrobial peptides. In this way, BaP-mediated upregulation of *Il22* might also contribute to the limitation of local infection and the development of systemic inflammation as well. The significance of IL-10 and IL-22 upon AhR activation for preventing septic course of S.E.-induced peritonitis has to be elucidated in further studies.

The promotor region of the *Il10* gene contains c-Maf binding sites and one XRE. Apetoh and colleagues could show that binding of activated AhR and c-Maf in the promotor region of the *Il10* gene leads to a stronger upregulation of *Il10* transcription in Treg cells [59,60]. These findings may explain the slightly upregulated *Il10* mRNA expression in splenocytes from BaP-exposed mice at day 3 p.i. in our model. On day 10 p.i., however, *Il10* transcription was no longer induced through BaP, neither in infected nor in non-infected animals, which coincided with the result of *Foxp3,* revealing no effect by any of both BaP doses in S.E.-infected animals.

Considering the anti-inflammatory impact of AhR activation causing a moderate or attenuated immune response to pathogens, it might be speculated that pathogenic bacteria have evolved mechanisms for AhR activation as an escape strategy for host adaptation to enable both their own survival and the survival of the host.

Another key marker and mediator of septic conditions is TNF-α [61]. The data about the influence of AhR on the regulation of TNF-α is quite controversial because the activation of AhR led to contrary effects in different models. Lecureur et al. reported on inducing effects of BaP in non-activated human macrophages [62], whereas in activated BMDCs, the secretion of TNF-α was impaired by different AhR ligands [43,63]. Moreover, we could show a decreased TNF-α secretion in response to S.E. antigen under BaP exposure of differentiated BMDMs [50]. However, when bone-marrow-derived myeloid precursor cells were differentiated to mature macrophages in the presence of BaP, TNF-α secretion is increased upon stimulation [51]. The present study revealed BaP-dependent suppressing effects on TNF-α expression, but in total, the results were not consistent. Particularly, the finding that lower TNF-α production in peritoneal inflammatory cells from S.E.-infected wild-type mice was AhR-dependent, due to higher TNF-α amounts in AhR-deficient mice, but independent on BaP exposure appears very interesting. It could be speculated that bacterial products from the salmonellae might suppress TNF-α production AhR-dependently via a newly-discovered mechanism. Very recently, Rosser et al. reported on AhR activation by microbiota-derived metabolites in regulatory B cells resulting in IL-10 production [64]. Hence, this mechanism of action has to be considered in our model and should be investigated in future studies.

To assess the impact of BaP on the humoral immune response, S.E.-specific IgG1 and IgG2c antibodies were analyzed. These Ig isotypes have been chosen to demonstrate the importance of IL-4 or IFN-γ, respectively. Although it has been reported in the literature that higher BaP doses may cause reduced antibody responses in vivo and in vitro [35,65], the production of S.E.-specific IgG1 and IgG2c antibodies were not impaired in our model, probably due to relatively low BaP doses (i.e., 20 ng/kg and 2 µg/kg). In contrast, the serum titers of both isotypes were equivalent or even higher than in vehicle controls and persisted up to day 90 p.i. An effective humoral immune response to S.E. was shown to be crucial for the eradication of salmonellae in this infection model [29]. Hence, the elevated IgG1 and IgG2c titers at day 90 p.i. are contradictory to the extracellular persistence of S.E. Extracellularly, salmonellae are highly sensitive to complement-mediated killing initiated by antibodies [34,38]. In this view, the finding of extracellular persistence of S.E.in BaP-exposed mice might indicate a deficient antibody-mediated complement-dependent killing. This assumption should be elucidated in follow-up studies.

## 5. Conclusions

In summary, the results of this study show that AhR activation through BaP exerts a significant impact on the course and outcome of septic peritonitis, and thus, highlight the key role of this receptor in immune cells. AhR activation is essential for the adjustment of the fragile balance between inflammatory and anti-inflammatory mechanisms that are triggered by the local or systemic cytokine milieu during the immune response. In the case of infections, AhR activation appears to be a double-edged sword. On the one hand, activation of the AhR by xenobiotic, nutritional, microbial or endogenous AhR ligands may attenuate or limit inflammatory responses and, thus, may prevent a septic course. On the other hand, this seems to hamper the eradication of the invaded pathogens, which, in turn, may promote the development of chronic infection and, as a consequence of that, chronic inflammatory diseases [19,58,66,67].

Overall, the data complete our knowledge on the complex area of AhR activation and the associated anti-inflammatory effects. This might be another milestone for the long way of developing AhR-directed therapeutic approaches for acute (e.g., SIRS/Sepsis) or chronic inflammatory diseases (e.g., chronic inflammatory bowel disease, rheumatoid arthritis, or allergies).

## Figures and Tables

**Figure 1 cells-11-00737-f001:**
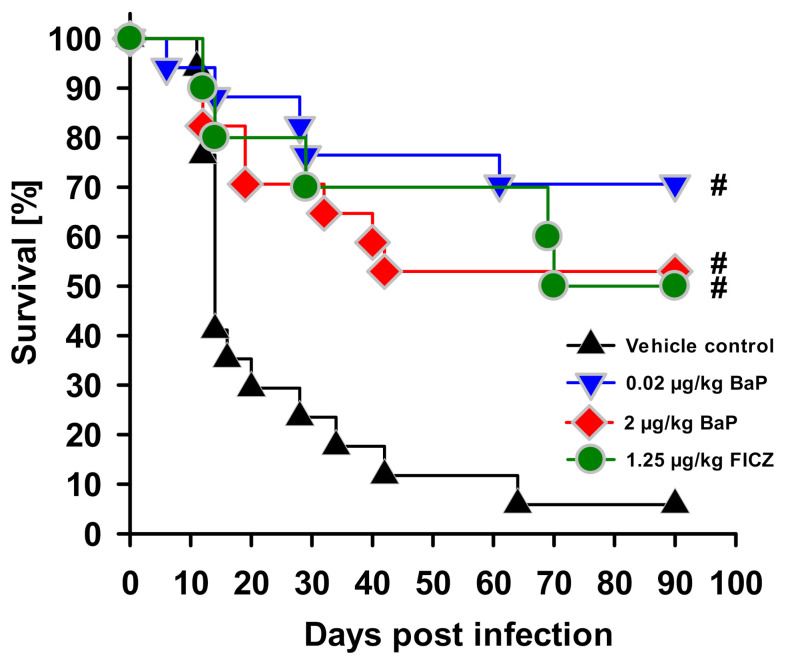
Effect of BaP on survival of mice during *S. enterica* infection. Mice were treated with BaP (0.02 µg/kg or 2 µg/kg body weight) or FICZ (1.25 µg/kg body weight) dissolved in corn oil (vehicle) or with the vehicle alone, via i.p. injection. Two days after first application of BaP, FICZ or vehicle, animals were systemically infected with salmonellae by i.p. injection of 5 × 10^7^ CFU of a live attenuated vaccine strain of *S. enterica* Serovar Enteritidis. Shown is the survival rate with the Kaplan–Meier method for one experiment (n = 10) over 90 days, and its representative of three independent experiments. Differences between treatment groups and the vehicle control were proved for statistical significance by the log-rank test (# *p* ≤ 0.01).

**Figure 2 cells-11-00737-f002:**
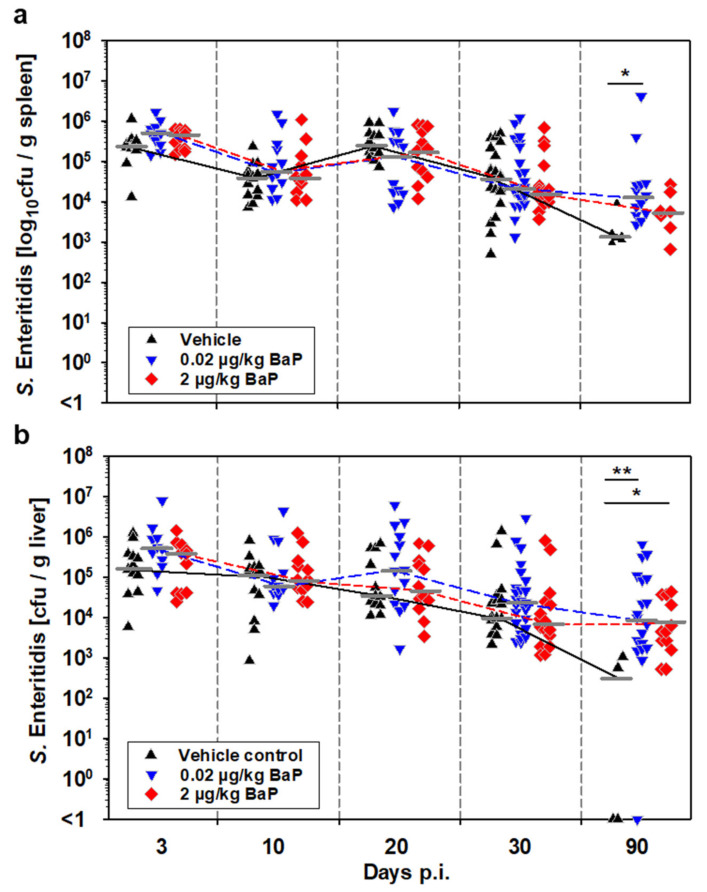
Effects of BaP on bacterial burden in total tissue homogenates of spleen (**a**) and liver (**b**) from S.E.-infected mice over the course of infection up to day 90 p.i. Mice were treated with BaP (0.02 µg/kg or 2 µg/kg body weight, dissolved in corn oil) or with the vehicle alone via i.p. injection. Two days after first application of BaP or vehicle, animals were systemically infected with salmonellae by i.p. injection of 5 × 10^7^ CFU of a live attenuated vaccine strain of *S. enterica* Serovar Enteritidis. Data derived from three independent experiments (n = 12–20), median values are indicated as short gray lines, which are connected by lines in the respective color for the three groups. Significant differences between treatment groups and the vehicle control group were estimated by Kruskal–Wallis one-way ANOVA on ranks followed by the Dunn’s test for multiple comparisons vs. control group (* *p* ≤ 0.05, ** *p* ≤ 0.01).

**Figure 3 cells-11-00737-f003:**
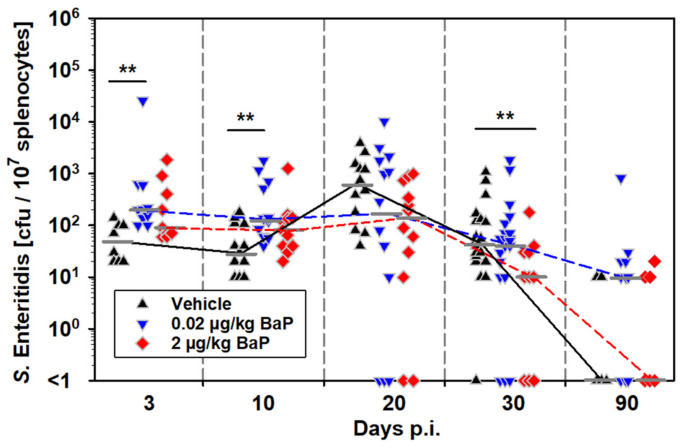
Effects of BaP on intracellular bacterial burden in white spleen cells (i.e., phagocytes) from S.E.-infected mice over the course of infection up to day 90 p.i. Mice were treated with BaP (0.02 µg/kg or 2 µg/kg body weight, dissolved in corn oil) or with the vehicle alone via i.p. injection. Two days after first application of BaP or vehicle, animals were systemically infected with salmonellae by i.p. injection of 5 × 10^7^ CFU of a live attenuated vaccine strain of *S. enterica* Serovar Enteritidis. Data derived from three independent experiments (n = 9–24), median values are indicated as short gray lines, which are connected by lines in the respective color for the three groups. Significant differences between treatment groups and the vehicle control group were estimated by Kruskal–Wallis one-way ANOVA on ranks followed by the Dunn’s test for multiple comparisons vs. control group (** *p* ≤ 0.01).

**Figure 4 cells-11-00737-f004:**
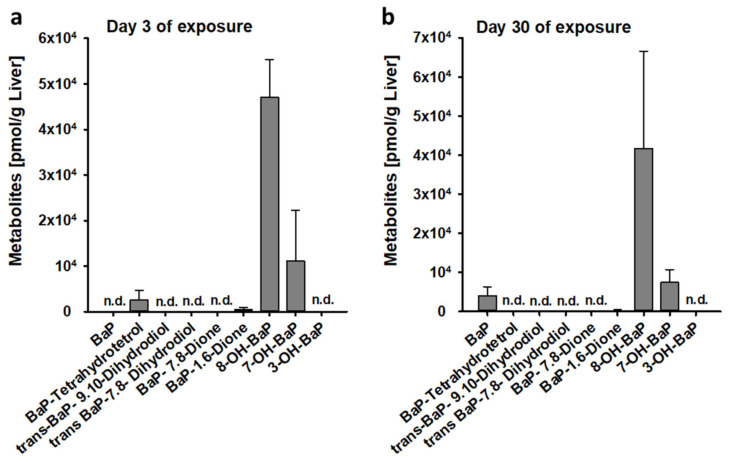
Bioavailability of BaP or BaP metabolites in the liver after 3-day (**a**) or 30-day exposure (**b**) of BaP. Mice were intraperitoneally (i.p.) injected with BaP (0.02 µg/kg body weight) dissolved in corn oil (vehicle). Three days or 30 days after BaP administration, animals were sacrificed, and liver tissue homogenates were analyzed for the yield of non-metabolized BaP and 8 prominent BaP metabolites by mass spectrometry. Data represent the mean values ± SEM, n = 3, n.d. not detectable.

**Figure 5 cells-11-00737-f005:**
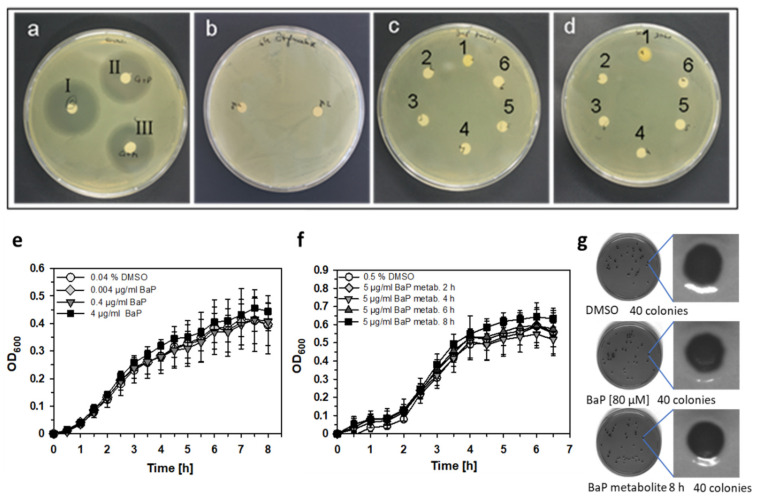
Growth behavior of *S. enterica* Serovar Enteritidis (S.E.) under exposure of BaP or BaP metabolites. S.E. was cultured for 24 h on LB agar. The following concentrations of BaP or BaP metabolites, transferred to filter paper discs on the agar plate, were tested. (**a**) Positive control (I) 100 µg/mL Gentamycin, (II) 50 µg/mL Gentamycin diluted in PBS and (III) 50 µg/mL Gentamycin diluted in complete RPMI1640 medium. (**b**) BaP metabolites—2 preparations; (**c**) BaP dissolved in corn oil—(1) vehicle control, (2) 80 mM, (3) 800 µM, (4) 80 µM, (5) 800 nM and (6) 8 nM; (**d**) BaP dissolved in DMSO—(1) vehicle control, (2) 40 mM, (3) 800 µM, (4) 80 µM, (5) 800 nM and (6) 8 nM. S.E. was incubated in LB medium for 9 h at 37 °C in a shaking incubator under exposure of indicated BaP (**e**) or BaP metabolite concentrations (**f**) without significant differences compared to the respective DMSO control. Shown are the proliferation curves, measured as the increase of optical density at a wavelength of 600 nm (OD_600_). Data represent the mean values ± SEM, n = 3. Differences between BaP- or BaP-metabolite treated groups and the solvent control DMSO were proved for statistical significance by Kruskal–Wallis one-way ANOVA or ANOVA on ranks (if normality and/or equal variance tests failed) followed by Holm–Sidak or Dunn’s post-hoc analysis, respectively. There were no significant differences. (**g**) Salmonellae were sequentially diluted in PBS from a frozen stock down to a theoretical yield of 50 colonies per 100 µL and plated on selective XLD agar. After 24-h incubation at 37 °C, colonies were counted and assessed in terms of the real number and the shape of the S.E. colonies.

**Figure 6 cells-11-00737-f006:**
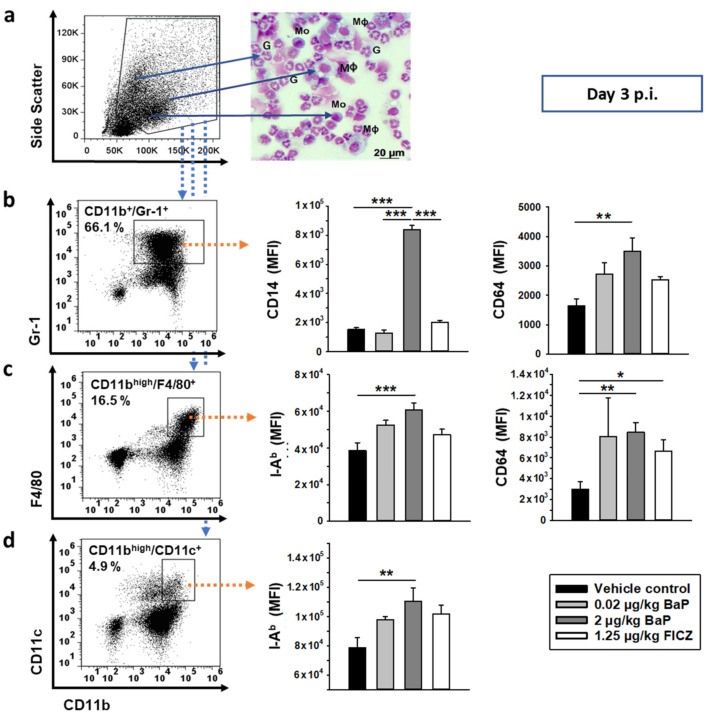
BaP-induced CD64, MHC class II and CD14 cell surface expression on peritoneal innate immune cells at day 3 of S.E. infection. Mice were treated with vehicle, 0.02 µg/kg or 2 µg/kg body weight BaP or 1.25 µg/kg body weight FICZ (both substances dissolved in corn oil) via i.p. injection. Two days after first application of BaP, FICZ or vehicle, animals were systemically infected with salmonellae by i.p. injection of 5 × 10^7^ CFU of a live attenuated vaccine strain of *S. enterica* Serovar Enteritidis (S.E.). (**a**) Gating strategy for acquisition and analyses of peritoneal myeloid cells. Pappenheim stain of cytospins from separated peritoneal cells revealed high proportions of neutrophilic granulocytes (G), monocytes (Mo) and macrophages (Mϕ). (**b**) CD11b^+^/Gr-1(Ly-6G/Ly-6C)^+^ peritoneal cells from S.E.-infected mice showed significantly increased expression of CD14 and CD64 in response to 2 µg/kg body weight BaP. (**c**) CD11b^high^/F4/80^+^ peritoneal cells from S.E.-infected mice showed increased expression of MHC class II (i.e., I-A^b^) and CD64 (FcγRI) in response to BaP (dose-dependently) or FICZ, respectively. (**d**) MHC class II (i.e., I-A^b^) expression was also significantly increased through 2 µg/kg body weight BaP on CD11b^high^/CD11c^+^ peritoneal cells from S.E.-infected mice. Results are shown as the mean of all mean fluorescence intensity values (MFI) ± SEM, n = 4–9. Differences between BaP- or FICZ-treated groups and the vehicle control were found statistically significant by Kruskal–Wallis one-way ANOVA followed by Holm–Sidak post-hoc analysis (* *p* < 0.05, ** *p* ≤ 0.005, *** *p* ≤ 0.001).

**Figure 7 cells-11-00737-f007:**
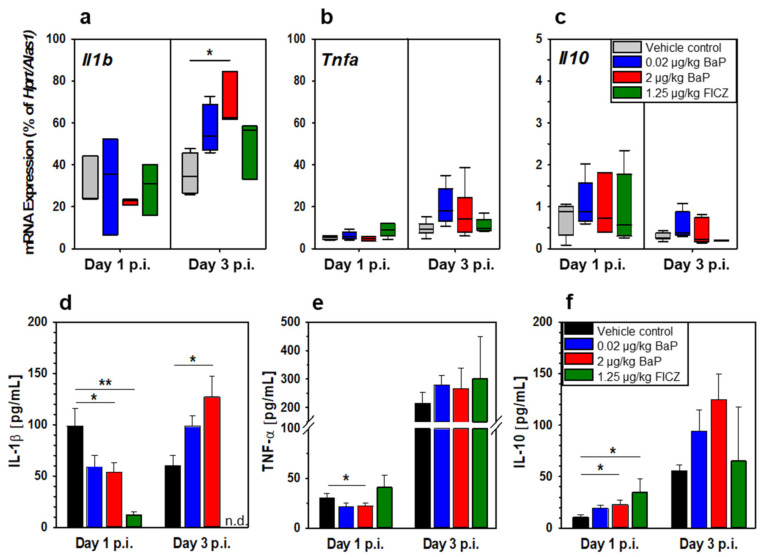
Comparison of BaP- and FICZ-induced effects on mRNA (**a**–**c**) and protein expression (**d**–**f**) of *Il1b*/IL-1β, *Tnfa*/TNF-α and *Il10*/IL-10 in peritoneal cells at days 1 and 3 post infection. Mice were treated with corn oil (vehicle control), 0.02 µg/kg or 2 µg/kg body weight BaP or 1.25 µg/kg body weight FICZ (both substances dissolved in corn oil) via i.p. injection. Two days after first application of BaP, FICZ or vehicle alone, animals were systemically infected with salmonellae by i.p. injection of 5 × 10^7^ CFU of a live attenuated vaccine strain of *S. enterica* Serovar Enteritidis (S.E.). Changes in gene transcription were assessed by qRT-PCR. Results are expressed as BaP- or FICZ-induced relative mRNA expression compared to vehicle control (upper panel, **a**–**c**). ∆*ct* values from test and control samples were normalized against the expression of two non-regulated housekeeping genes, *Alas1* and *Hprt*, and expressed in % of the transcription rate of the housekeeping genes. Data are shown as boxplots indicating the median and 25% and 75% confidence intervals, n = 5–6 mice per group. Protein concentration was analyzed in peritoneal cavity fluid by cytometric bead array (lower panel, **d**–**f**). Plotted data represent mean ± SD, n = 5–14 mice per group. Differences between BaP- or FICZ-treated groups and the vehicle control were proved for statistical significance by Kruskal–Wallis one-way ANOVA or ANOVA on ranks followed by Holm–Sidak or Dunn’s post-hoc analysis (* *p* ≤ 0.05, ** *p* ≤ 0.01). n.d. not determined.

**Figure 8 cells-11-00737-f008:**
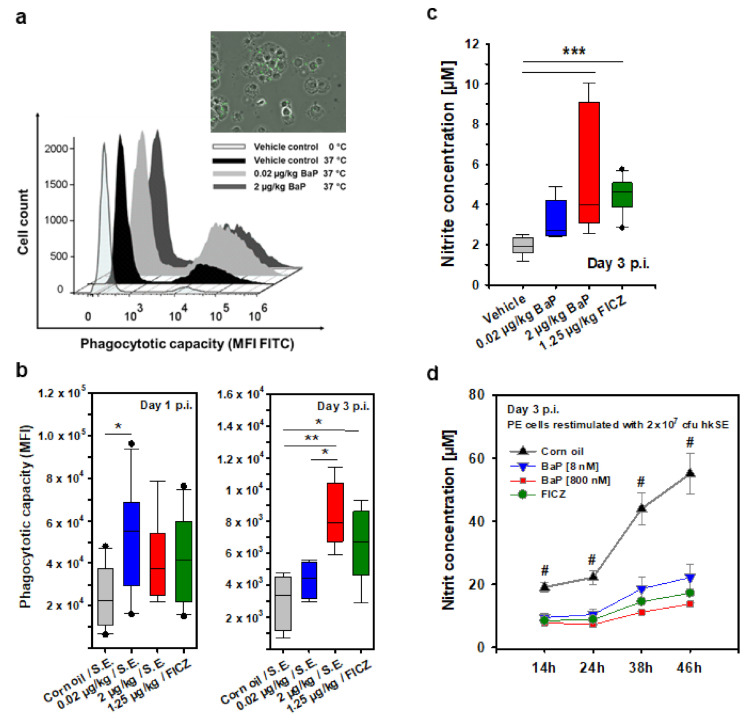
Effect of BaP on phagocytosis of FITC-labeled heat-killed salmonellae (hk S.E.) by peritoneal exudate cells derived from S.E.-infected mice at days 1 and 3 p.i. Peritoneal cavity phagocytes were isolated by lavage and cultured with FITC-labeled hk S.E. for 1 h. Phagocytotic capacity was assessed by flow cytometry. Mice were treated with vehicle, 0.02 µg/kg or 2 µg/kg body weight BaP or 1.25 µg/kg body weight FICZ (both substances dissolved in corn oil) via i.p. injection. Two days after first application of BaP, FICZ or vehicle, animals were systemically infected with salmonellae by i.p. injection of 5 × 10^7^ CFU of a live attenuated vaccine strain of *S. enterica* Serovar Enteritidis (S.E.). (**a**) The flow cytometric histograms and the fluorescence photomicrograph (merge of phase contrast picture of peritoneal exudate (PE) macrophages and immunofluorescence staining of salmonellae using an S.E.-specific polyclonal antibody) derived from one animal are demonstrated exemplarily. (**b**) Data are shown as boxplots indicating the median and 25% and 75% confidence intervals, n = 5 mice per group. Differences between BaP- or FICZ-treated groups and the vehicle control were tested for statistical significance by one-way ANOVA followed by post-hoc analysis with the Holm–Sidak method (* *p* ≤ 0.05, ** *p* ≤ 0.01). (**c**) Nitrite in the supernatant of peritoneal lavage was measured using the Griess reaction. Data are shown as boxplots indicating the median and 25% and 75% confidence intervals, n = 4 mice per group. Differences between BaP- or FICZ-treated groups and the vehicle control were tested for statistical significance by one-way ANOVA followed by post-hoc analysis with the Holm–Sidak method or by Kruskal–Wallis one-way ANOVA on ranks (if normality and/or equal variance test have not passed) followed by Dunn’s post-hoc analysis (*** *p* ≤ 0.001). (**d**) Nitrite concentration in supernatants of PE cells at different time points after ex-vivo stimulation with 2 × 10^7^ CFU hk S.E. Individual time points of different treatments were found for statistical significance by Kruskal–Wallis one-way ANOVA on ranks followed by the Tukey test for pairwise comparisons on mean ranks of treatment groups (# *p* ≤ 0.004).

**Figure 9 cells-11-00737-f009:**
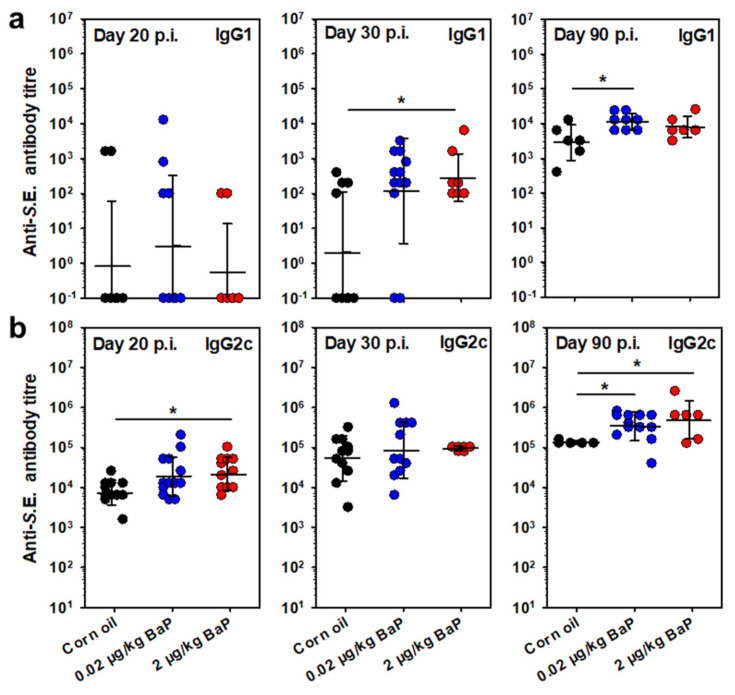
Effects of BaP on the *S. enterica*-specific antibody response over the course of infection (analyzed at days 20, 30 and 90 p.i.). Mice were treated with vehicle (corn oil), 0.02 µg/kg or 2 µg/kg body weight BaP (dissolved in corn oil) via i.p. injection. Two days after first application of BaP or vehicle, animals were systemically infected with salmonellae by i.p. injection of 5 × 10^7^ CFU of a live attenuated vaccine strain of *S. enterica* Serovar Enteritidis (S.E.). On days 20, 30 and 90 p.i. sera were generated, and the titers of S.E.-specific IgG1 (**a**) and IgG2c antibodies (**b**) were analyzed using an in-house ELISA. Dots in the vertical point plot represent the titer values from individual mice. In addition, the mean values ± SEM are indicated. Differences between BaP-treated groups and the vehicle control were tested for statistical significance by Kruskal–Wallis one-way ANOVA or ANOVA on ranks (if normality and/or equal variance tests failed) followed by post-hoc analysis with the Holm–Sidak method or the Dunn’s test, respectively (* *p* ≤ 0.05).

**Figure 10 cells-11-00737-f010:**
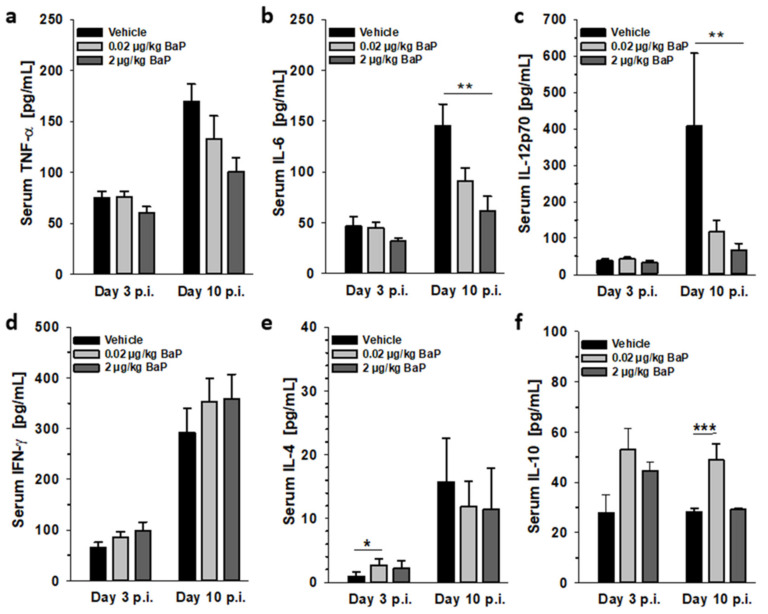
Effects of BaP on the systemic cytokine response (spleen) at days 3 and 10 p.i. Mice were treated with vehicle (corn oil), 0.02 µg/kg or 2 µg/kg body weight BaP (dissolved in corn oil) via i.p. injection. Two days after first application of BaP or vehicle, animals were systemically infected with salmonellae by i.p. injection of 5 × 10^7^ CFU of a live attenuated vaccine strain of *S. enterica* Serovar Enteritidis (S.E.). On days 3 and 10 p.i. sera were generated, and concentrations of relevant proinflammatory (i.e., TNF-α, IL-6, IL-12, IFN-γ (**a**–**d**)) and anti-inflammatory cytokines (i.e., IL-4, IL-10; (**e**,**f**) were analyzed by cytometric bead array (CBA)). Plotted data represent the mean values ± SD. Differences between BaP-treated groups and the vehicle control were tested for statistical significance by Kruskal–Wallis one-way ANOVA or ANOVA on ranks (if normality and/or equal variance tests failed) followed by post-hoc analysis with the Holm–Sidak method or the Dunn’s test, respectively (* *p* ≤ 0.05, ** *p* ≤ 0.01, *** *p* ≤ 0.001).

**Figure 11 cells-11-00737-f011:**
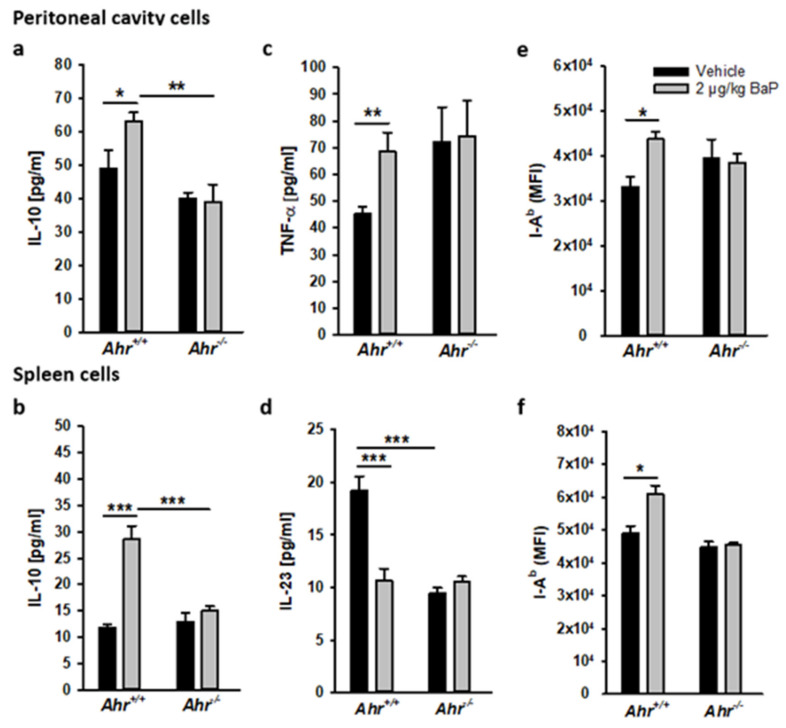
AhR-dependent effects of BaP on the local and systemic cytokine response and MHC class II surface expression by peritoneal cavity cells (upper panel) or spleen cells (lower panel), **respectively.** *Ahr* wild-type (*Ahr^+/+^*) and *Ahr*-deficient mice (*Ahr^−/−^*) were intraperitoneally injected with 2 µg/kg BaP or vehicle (corn oil). After 20 h of BaP exposure in vivo, white cells from peritoneal cavity and spleen were isolated, washed three times in PBS and cultured in RPMI1640 medium (10% FCS) at 37 °C, 5% CO_2_, 96% humidity in the presence of 2 × 10^7^ CFU hkS.E. for 20 h ex vivo. Then, culture supernatants were harvested for cytokine analysis. Concentrations of IL-10 (**a**,**b**), TNF-α (**c**) and IL-23 (**d**) were determined by ELISA. Cells were harvested and immunophenotyped for CD11b, F4/80 and MHC II (I-A^b^) expression. Shown is the mean fluorescence intensity (MFI) of I-A^b^ staining on CD11b^high^/F4/80^+^ cells (**e**,**f**). Plotted data represent the mean values ± SEM (n = 4). Differences between BaP treatment and vehicle control or between *Ahr^+/+^* and *Ahr^−/−^* mice were tested for statistical significance by Student’s *t*-test if the normality and equal variance test passed or by the Mann–Whitney rank sum test (*u*-test) if the normality and/or equal variance test failed (* *p* ≤ 0.05, ** *p* ≤ 0.01, *** *p* ≤ 0.001).

**Figure 12 cells-11-00737-f012:**
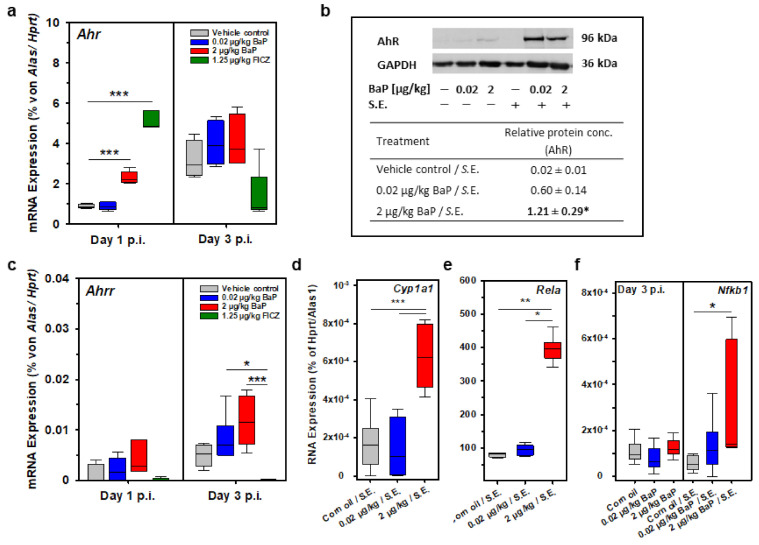
BaP- or FICZ-induced mRNA and protein expression of AhR target genes in spleen cells from *S. enterica*-infected C57BL/6 mice at days 1 and 3 p.i. Mice were treated 3 times per week with BaP (0.02 µg/kg or 2 µg/kg body weight, dissolved in corn oil) or with the vehicle alone, via i.p. injection. Two days after first application of BaP or vehicle, animals were systemically infected with salmonellae by i.p. injection of 5 × 10^7^ CFU of a live attenuated vaccine strain of *S. enterica* Serovar Enteritidis (S.E.). (**a**) Significant *AhR* mRNA expression was observed at day 1 p.i. in response to 2 µg/kg BaP and 1.25 µg/kg FICZ. On day 3 p.i. *Ahrr* (**c**) and *Cyp1a1* transcription (**d**) were found to be induced in spleen cells from S.E.-infected mice under exposure with BaP. Moreover, *Rela* (**e**) and *Nfkb1* (**f**) were upregulated through BaP exposure in S.E.-infected mice at day 3 p.i. Data are shown as mean ± SEM, n = 5 mice per group. Gene expression was determined as the relative expression in relation to two different reference genes (*Alas*, *Hprt*), which have been defined as 100% expression. Differences between BaP-treated groups and the vehicle control were proved for statistical significance by ANOVA followed by post-hoc analysis with the Holm–Sidak method (* *p* ≤ 0.05, ** *p* ≤ 0.01, *** *p* ≤ 0.001). (**b**) AhR protein expression was detected by Western blot and normalized to the reference protein GAPDH. BaP treatment caused induction of AhR in spleen cells from S.E.-infected mice but only to a weak extent in spleen cells from non-infected mice. The Western blot from one animal was shown exemplarily. All Western blot results were analyzed by densitometry; the relative AhR protein concentration is shown as the ratio of AhR signal/GAPDH signal (reference protein). Values represent mean ± SEM, n = 5 mice per group. Differences between BaP-treated groups and the vehicle control were proved for statistical significance by ANOVA followed by post-hoc analysis with the Holm–Sidak method (* *p* ≤ 0.05).

**Figure 13 cells-11-00737-f013:**
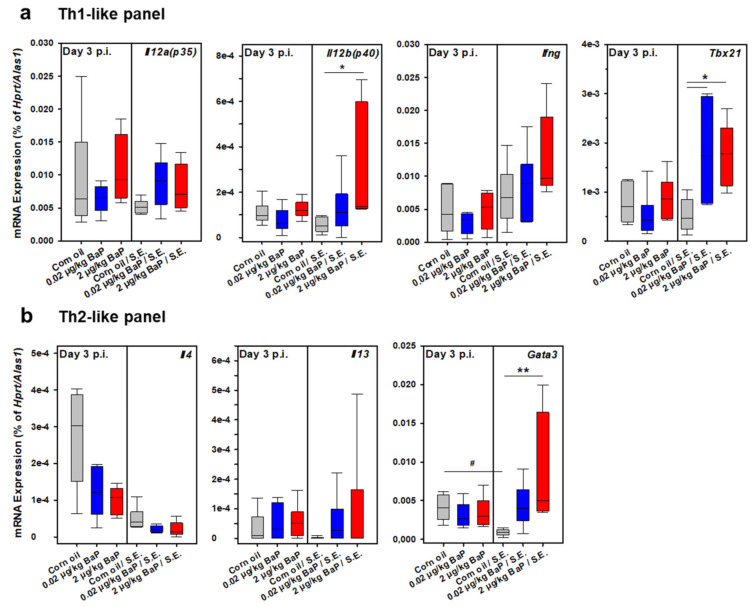
Effects of BaP on cytokine mRNA expression in the spleen on day 3 post-infection (p.i.). Mice were treated with vehicle, 0.02 µg/kg or 2 µg/kg body weight BaP (dissolved in corn oil) via i.p. injection. Two days after first application of BaP or vehicle, animals were systemically infected with salmonellae by i.p. injection of 5 × 10^7^ CFU of a live attenuated vaccine strain of *S. enterica* Serovar Enteritidis (S.E.). Changes in gene transcription were assessed by qRT-PCR and classified into the following expression patterns: (**a**) Th1-like panel (*Il12a, Il12b, Ifng, Tbx21*); (**b**) Th2-like panel (*Il4, Il13, Gata3*); (**c**) ILC3/Th17/Th22-like panel (*Il17a, Rorc, Il22, Ahr*); (**d**) Treg-like panel (*Il6, Tgfb1, Il10, Fop3*); (**e**) inflammatory panel (*Il1b, Tnf, Il18*). Results are expressed as BaP-induced relative mRNA expression compared to vehicle-treated mice. ∆*ct* values from test and control samples were normalized against the expression of two non-regulated housekeeping genes, *Alas1* and *Hprt,* and expressed in % of the transcription rate of the housekeeping genes. Data are shown as boxplots indicating the median and 25% and 75% confidence intervals, n = 5–6 mice per group. Differences between BaP-treated groups and the vehicle control were tested for statistical significance by one-way ANOVA followed by post-hoc analysis with the Holm–Sidak method or by Kruskal–Wallis one-way ANOVA on ranks (if normality and/or equal variance test have not passed) followed by Dunn’s or Tukey’s post-hoc analysis (* *p* ≤ 0.05, ** *p* ≤ 0.01). Differences between the respective groups of non-infected and S.E.-infected mice were tested in the same way (# *p* ≤ 0.05, ## *p* ≤ 0.01, ### *p* ≤ 0.001).

**Figure 14 cells-11-00737-f014:**
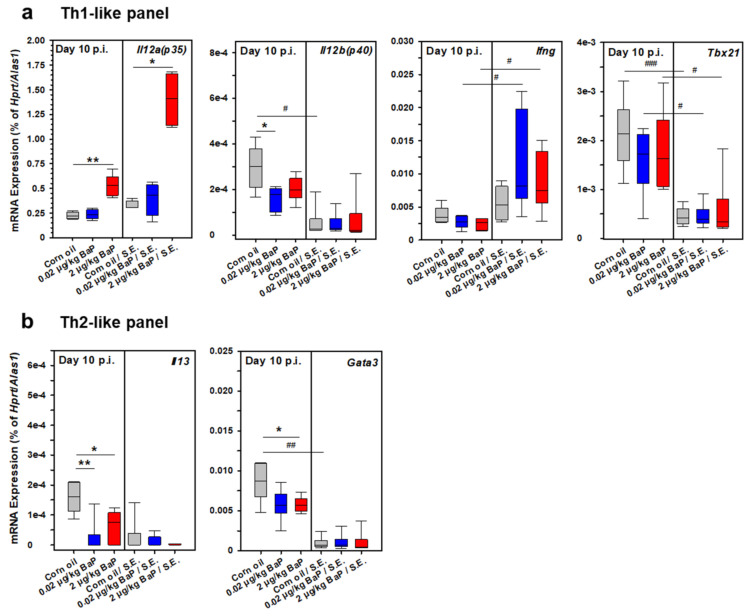
Effects of BaP on mRNA expression of selected cytokines in the spleen on day 10 post-infection (p.i.). Mice were treated with vehicle, 0.02 µg/kg or 2 µg/kg body weight BaP (dissolved in corn oil) via i.p. injection. Two days after first application of BaP or vehicle, animals were systemically infected with salmonellae by i.p. injection of 5 × 10^7^ CFU of a live attenuated vaccine strain of *S. enterica* Serovar Enteritidis (S.E.). Changes in gene transcription were assessed by qRT-PCR and classified into the following expression patterns: (**a**) Th1-like panel (*Il12a, Il12b, Ifng, Tbx21*); (**b**) Th2-like panel (*Il4, Il13, Gata3*); (**c**) ILC3/Th17/Th22-like panel (*Il17a, Rorc, Il22, Ahr*); (**d**) Treg-like panel (*Il6, Tgfb1, Il10, Fop3*); (**e**) inflammatory panel (*Il1b, Tnf, Il18*). Results are expressed as BaP-induced relative mRNA expression compared to vehicle-treated mice. ∆*ct* values from test and control samples were normalized against the expression of two non-regulated housekeeping genes, *Alas1* and *Hprt,* and expressed in % of the transcription rate of the housekeeping genes. Data are shown as boxplots indicating the median, 25% and 75% confidence intervals, n = 5–6 mice per group. Differences between BaP-treated groups and the vehicle control were tested for statistical significance by Kruskal–Wallis one-way ANOVA followed by post-hoc analysis with the Holm–Sidak method or ANOVA on ranks (if normality and/or equal variance tests failed) followed by Dunn’s or Tukey’s post-hoc analysis (* *p* ≤ 0.05, ** *p* ≤ 0.01, *** *p* ≤ 0.001). Differences between the respective groups of non-infected and S.E.-infected mice were tested in the same way (# *p* ≤ 0.05, ## *p* ≤ 0.01, ### *p* ≤ 0.001).

**Table 1 cells-11-00737-t001:** Primers and corresponding ULP probes used for quantitative *real-time* RT-PCR.

Gene	Forward Primer (5′-3′)	Reverse Primer (5′-3′)	UPL Probe
*Ahr*	tgcacaaggagtggacga	aggaagctggtctggggtat	27
*Ahrr*	cagggtaaagagcttcttccaa	ggggaaccctctgtatgagtg	19
*Alas1*	ccctccagccaatgagaa	gtgccatctgggactcgt	40
*Arnt*	tgcctcatctggtactgctg	tgtcctgtggtctgtccagt	108
*cJun*	ccagaagatggtgtggtgttt	ctgaccctctccccttgc	11
*Cyp1a1* *Cyp1a2*	tccctccttacagcccaag cctggactgactcccacaac	acgaaggctggaagtccata cgccatctgtaccactgaag	97 19
*Foxp3*	agaagctgggagctatgcag	gctacgatgcagcaagagc	20
*Gata3*	ttatcaagcccaagcgaag	tggtggtggtctgacagttc	108
*G6pdh*	ccagcccttcccctatgtat	gtactggaagcccactctcct	34
*Ef2*	gtagatgccacccaccactt	gctgcctgtatgccagtgt	47
*Gapdh*	gagccaaacgggtcatca	catatttctcgtggttcacacc	29
*Hprt*	tcctcctcagaccgctttt	cctggttcatcatcgctaatc	95
*Il1b*	ttgacggaccccaaaagat	gaagctggatgctctcatctg	26
*Il4*	catcggcattttgaacgag	tctgtggtgttcttcgttgc	02
*Il6*	gctaccaaactggatataatcagga	ccaggtagctatggtactccagaa	06
*Il10*	gctcctagagctgcggact	tgttgtccagctggtccttt	41
*Il12a*	atcacaaccatcagcagatca	cgccattatgattcagagactg	49
*Il12b*	cgcagcaaagcaaggtaagt	cctctagatgcagggagttagc	103
*Il17a*	gattttcagcaaggaatgtgg	cattgtggagggcagacaat	34
*Il18*	catgtacaaagacagtgaagtaagagg	tttcaggtggatccatttcc	76
*Il22*	tgacgaccagaacatccaga	cgccttgatctctccactct	94
*Il23a*	ctgttgccctgggtcact	agcccagtcaggactgctac	76
*Ifng*	atctggaggaactggcaaaa	ttcaagacttcaaagagtctgaggta	21
*iNos*	gggctgtcacggagatca	ccatgatggtcacattctgc	76
*Nfkb1*	cactgctcaggtccactgtc	ctgtcactatcccggagttca	69
*Rela*	cccagaccgcagtatccat	gctccaggtctcgcttctt	47
*Rorc*	acctcttttcacgggagga	tcccacatctcccacattg	6
*Tbx21*	tcaaccagcaccagacagag	aaacatcctgtaatggcttgtg	19
*Tgfb*	tggagcaacatgtggaactc	gtcagcagccggttacca	72
*Tnf*	tcttctcattcctgcttgtgg	ggtctgggccatagaactga	49

**Table 2 cells-11-00737-t002:** Influence of BaP on the distribution of selected innate immune cells. Shown are the proportions of positive cells (%) in the white splenocyte fraction from S.E.-infected mice at days 1 and 3 p.i. There were no statistically significant differences between BaP-exposed groups and the vehicle control (proved by ANOVA).

Cell Population	CD11b^high^/F4/80^+^	CD11b^+^/Gr1^+^	CD11b^high^/CD11c^+^	CD3^−^/NK1.1^+^/NKp46^+^
**Day 1 p.i.**				
Vehicle	4.3 ± 0.5	3.4 ± 1.2	2.9 ± 0.5	0.9 ± 0.1
0.02 µg/kg BaP	3.8 ± 0.3	3.3 ± 1.3	2.8 ± 0.4	1.1 ± 0.1
2 µg/kg BaP	4.9 ± 0.4	3.6 ± 1.4	2.7 ± 0.6	1.1 ± 0.3
**Day 3 p.i.**				
Vehicle	3.5 ± 0.5	11.7 ± 2.9	3.4 ± 0.4	3.0 ± 0.4
0.02 µg/kg BaP	3.8 ± 0.5	12.2 ± 2.9	3.4 ± 0.4	2.3 ± 0.7
2 µg/kg BaP	3.8 ± 0.4	14.0 ± 2.6	3.9 ± 1.2	2.5 ± 0.4

**Table 3 cells-11-00737-t003:** Effect of BaP treatment on the expression level of MHC class II (I-A^b^), CD64 (FcγRI), CD80 (B-7.1) and CD86 (B-7.2) and CD14 on different splenic innate immune cell populations from S.E.-infected mice at days 1 and 3 p.i. Shown are the values for mean fluorescence intensity (MFI), which represent a relative correlate to the number of molecules per cell (mean ± SEM, n = 4–6).

Cell Population	CD11b^high^/F4/80^+^			
Functional marker	MHC-II (I-A^b^)	CD64	CD86	CD80
**Day 1 p.i.**				
Vehicle	34,450 ± 2029	4230 ± 234	2928 ± 232	2644 ± 137
0.02 µg/kg bw BaP	**48,800 ± 4300 ***	4535 ± 341	2602 ± 154	2807 ± 112
2 µg/kg bw BaP	40,950 ± 2157	**5171 ± 278 ***	2883 ± 186	3073 ± 219
**Day 3 p.i.**				
Vehicle	14,025 ± 1202	3181 ± 114	1620 ± 37	1976 ± 112
0.02 µg/kg bw BaP	14,550 ± 639	**3700 ± 147 ***	1741 ± 43	1790 ± 102
2 µg/kg bw BaP	16,233 ± 1876	3584 ± 249	1670 ± 76	1987 ± 85
**Cell population**	**CD11b^+^/Gr-1^+^**		**CD11b^high^/CD11c^+^**	
Functional marker	CD14	CD64	MHC-II (I-A^b^)	
**Day 1 p.i.**				
Vehicle	5937 ± 812	8734 ± 403	52,000 ± 2.441	
0.02 µg/kg bw BaP	7097 ± 652	9839 ± 990	56,600 ± 2.536	
2 µg/kg bw BaP	**7956 ± 270 ***	**11,028 ± 897 ***	58,420 ± 2.904	
**Day 3 p.i.**				
Vehicle	2593 ± 377	2960 ± 427	46,828 ± 1390	
0.02 µg/kg bw BaP	2818 ± 213	2890 ± 293	**51,750 ± 1550 ***	
2 µg/kg bw BaP	2926 ± 145	2761 ± 319	**50,675 ± 677 ***	

* significant in comparison to vehicle control (*p* ≤ 0.05, ANOVA/Student–Newman–Keuls post-hoc test).

**Table 4 cells-11-00737-t004:** Effect of BaP treatment on the phagocytosis of FITC-labeled heat-killed salmonellae (hk S.E.) by peritoneal cavity cells derived from S.E.-infected mice at day 3 p.i. Peritoneal cavity phagocytes were isolated by lavage and cultured with FITC-labeled hk S.E. for 1 h. Phagocytotic capacity was assessed by flow cytometry.

	Phagocytotic Capacity	
	MFI	Positive Cells (%)
Vehicle	27,211 ± 15,613	23.57 ± 7.62
0.02 µg/kg bw BaP	**52,120 ± 25,018 ***	**37.10 ± 7.77 ***
2 µg/kg bw BaP	35,500 ± 19,465	**37.83 ± 9.21 ***

* significant in comparison to vehicle control (n = 4, *p* ≤ 0.01).

## Data Availability

All data generated or analyzed during this study are included in this published article or are available as supporting material. The mouse infection model used in this study can be made available for basic as well as applied research and for preclinical drug development (GLP level).

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
