# Peer review of "Aryl Hydrocarbon Receptor Activation by Benzo[a]pyrene Prevents Development of Septic Shock and Fatal Outcome in a Mouse Model of Systemic Salmonella enterica Infection"

_cells, 2022, doi:10.3390/cells11040737_

Round 1
Reviewer 1 Report
This paper reports a set of studies characterizing the effects of benzo[a]pyrene on aspects of the immune response to Salmonella enterica in mice. The experimental work appears to be very well done, and findings are interesting. The paper contains a lot of data, and the amount of work reflected in this paper is very much appreciated. Yet, the work is not presented in a cohesive manner, and while the authors’ enthusiasm for the topic is apparent, it seems that much of the data are over-interpreted. Suggestions to improve the paper are provided. As an overarching comment, the paper is not very cohesive, and has too many different threads that, in the current form, gel together or support one another in a compelling or clear manner.
Major issues to address, some of which may require additional experiments.
- Overall, the authors found that treating mice with BaP or FICZ reduced mortality due to enterica infection caused by intraperitoneal injection of bacteria. This is a very intriguing observation, and a great first figure. As the authors point out, this observation is novel but not inconsistent with prior reports of AHR activation or absence affecting host defenses against other bacteria. Yet, their data would be even more robust if global Ahr-/- were treated with BaP or FICZ and infected, to confirm AHR specificity of these effects of BaP and FICZ. It is possible that improved survival is due to non-AHR mediated events, as BaP is metabolized to several biological active intermediates, and FICZ has a short in vivo half-life. It is also possible that these effects of BaP are AHR-mediated, but caused by downstream metabolites of BaP (which could affect either the host immune response or the bacteria), and this was not considered. It is clear that Ahr-/- mice were available; however, the experiments performed using them (e.g., measuring IL-10, TNF and/or IL-23 in supernatants from ex vivo cultured peritoneal and spleen cells, and the relative level of MHCII on some of these cells) are not compelling with regards to rationale or findings, and did not accomplish the authors’ goal of demonstrating AHR-dependence of the major observation.
- The authors denote in vivo doses of BaP as “expected subtoxic,” “expected lower toxic,” and “expected toxic range,” but provide neither data nor citations to support these statements. Please provide evidence to support these descriptors. It would also be beneficial to include data (e.g., Cyp1a1 or another validated AHR target gene) showing dosing with BaP or FICZ every 3rd day maintained AHR activation throughout the 90 day time period.
- While I am confident that the experiments were performed with care, in many instances the data presented are over interpreted. For example, the findings in Figure 2 are solid assessments of bacterial burden over time; yet, the conclusions that more bacteria are surviving in splenic phagocytes, and that the higher bacterial loads on day 90 post infection reflect extracellular bacteria need to be removed or more experimental work performed and included to substantiate these statements. (Given the amount of work involved, this reviewer is recommending removal of Figure 3, revising the text, and presenting a simpler and more circumspect interpretation of the findings presented in Figure 2).
- Fig 4 and Tables 2 and 3: the phenotypic markers and gating strategies used to define cell populations are a little bit unusual. For example, while F4/80 is one of the most commonly used markers of mouse macrophages, not all monocytes express F4/80, and are generally distinguished from other leukocytes by the expression of Ly6C. A more straightforward definition of cell types and gating strategies to identify them would help make these data easier for others to appreciate. Likewise, the definition of DCs is hard to understand, and the rational for focusing in on MHCII levels is unclear. (i.e., some dendritic cells express CD11b, whereas other DC subsets do not. Therefore, the initial examination of DCs based on co-expression of CD11b and CD11c, with follow-up examination of relative level of MHCII (based on a modest shifts in MFI) is odd and the biological rationale is not clear). Moreover, to interpret impact of a chemical on different populations of immune cells, in addition to percentage, the number should also be considered. Finally, given that a single point in time is examined, it is possibly an overstatement to frame these data as providing information about recruitment of cells to site of infection.
- Table 4 and Figure 6: BaP appeared to affect the ability of peritoneal cells to take up enterica. Given that the authors used flow cytometry to measure FITC labeled S. enterica, it is unfortunate that they did not parse out the specific cell type(s) taking up the bacteria and that are affected by BaP. This refinement of their analyses could provide more nuanced information, and might help integrate some of the other data reported in the paper.
- Cytokines in serum were measured to examine whether BaP affects systemic immune response to enterica. Overall, the impact was modest, but generally consistent in that higher levels of these cytokines 10 days relative to 3 days post infection. BaP reduced IL12 (p70) levels, had no effect on IFNg, and a modest albeit non-dose-dependent impact on IL-10. Throughout this section, the authors over interpret the data, referring to trends that are not supported by statistical tests, nor in some instances observable to this reviewer. These statements should be removed.
- It is very logical to also examine antibody levels, given that many AHR ligands, including BaP, have been shown to affect humoral immune responses to other antigens. However, the findings from these ELISAs are a little bit confusing as presented. It is a bit surprising that enterica antibody levels were either not affected or increased in BaP-treated mice. Given that AHR activation generally represses antibody responses, this seems like an observation that requires more experimental follow up. Also, at a technical level, there are several things to consider. For instance, for IgG2c the statistical difference from corn oil control may be driven by the fact that there was essentially no variation in the control samples. This may reflect that the intervals of dilations needs to be narrowed. Given the novelty of this observation, it would benefit from using a secondary method to confirm this potentially very interesting finding.
- The analysis of gene expression in bulk spleen cells 3- and 10-days post infection was used as a means to explore possible mechanisms by which BaP could prevent fatal outcome of peritoneal S. enterica infection. This is logical approach to identify testable mechanistic hypotheses, but as presented, the data are heavily over-interpreted and ignore that, for many of the genes examined, more than one cell type expresses them. It was also unclear whether all of the data were provided (and if not then they need to be). Without follow up experiments, the authors need to be much more circumspect in the interpretation of these data, and eliminate text in which they describe trends.
- The pivot to in vitro experiments with bone marrow derived macrophages treated with either BaP or I3C does not fit well in the paper. This seems like a new line of inquiry. It is recommended that the authors remove section 3.9, Tables 5-7, and Figures 12 and 13 from the paper. As a side note, the absence of an effect of BaP or I3C on Cyp genes is mentioned, but not shown. This is moderately concerning, and reinforces the idea that the authors need to confirm that the overall effect of BaP on mortality (in vivo) is AHR-dependent. This also suggests that they need to perform additional experiments with the bone marrow derived macrophage system.
- These current findings need to be integrated with published literature on BaP immunotoxicity. There is quite an extensive literature, published by different groups of authors, yet this was absent in the current paper. Inclusion is necessary, and would enrich the Discussion. Also, the authors probably need to consider that metabolites of BaP can impact immune cell responses (and cellular function in general) and weave this into their thinking.
- The authors indicate ‘data not shown’ in several instances. They either need to include these data or not mention them, because there is no way for anyone to independently consider these findings.
Minor
- The Introduction is very long, and contains information that would be well suited to a review article, but presents details that are less clearly applicable or necessary for this research article. Information should be condensed and details that are extraneous removed (or moved to the Discussion). In particular, information that is included here but that is not part of experimental work in this paper, such as ‘canonical and non-canonical’ AHR signaling pathways are distracting in the Introduction and, more to the point, are not part of the experimental question or approach; therefore, it is unclear why they are mentioned in the Introduction.
- While the authors’ transparency as to which statistical tests were used is welcome, there was a bit of mixing of tests used even within one data set, blending t-tests and ANOVAs. Given the study design, a Student’s t-test in not really an appropriate tool to use. Also, within one dataset, using a variety of tests to denote p<0.05 is not the most ideal or correct statistical approach.
- Throughout the paper there are minor issues related to English syntax and grammar that, if corrected, will enhance readers’ ability to understand the authors’ points. As just one example, page 2 lines 44-46.
- The nitrite assessment is presented with rationale, but the data are confusing. Specifically, panels 6c and 6d do not seem to align well. It is also not clear how this information adds to the paper as a whole as it is quite speculative assessment of a non-specific response.
- The strain of Ahr-/- mice used is not reported using the name of the strain.
- There is no rationale provided as to why the authors used FICZ for in vivo studies, but I3C for in vitro Ideally, the same agonist should have been used. (Although this reviewer recommends removing the bone marrow derived macrophage studies altogether).
Reviewer 2 Report
Fueldner et al reports interesting study, where they observed that benzo-a-pyrene in sub-toxic doses prevents systemic inflammatory syndrome induced by salmonella infection in mice. Paradoxically, whereas inflammation was attenuated by BaP, the amount of pathogen in mice was increased. The mechanism for BaP effects comprised both canonical and non-canonical aryl hydrocarbon receptor AhR signaling. Study is presented in clear manner, experimental set up is well designed and the data are interpreted properly. I do not have any major critics, but considering following conceptual point would significantly improve the positioning the paper in broader context:
Whereas applied BaP is prototypical xenobiotic ligand and agonist of the AhR, human (and also mice) body is exposed to myriad of circulating and tissue-specific endogenous and microbial AhR ligands. Proper portfolio of these ligands, both in sites and time, leads to desirable and dynamic activation of the AhR, which in turn ensures normal physiological functions. If under-activated (microbiota or metabolic disbalance) or over-activated (by xenobiotics), the AhR becomes a culprit of patho-physiological effects, occurring in multiple organs (e.g. concept described here: Trends Pharmacol Sci 2020, 41(12):900-908). Could the authors dedicate a paragraph (in intro, discussion) to the orchestrated interaction of endogenous, microbial and xenobiotic AhR ligands, in the context of bacterial infections?
Round 2
Reviewer 1 Report
This paper contains a lot of interesting observations, which if framed in a more circumspect manner, would be a nice addition to the field. The additional data that have been added are appreciated. Yet, some of the major concerns expressed remain. For instance, while I am empathetic that the authors wish to use some data that are absent in this paper for a different paper, they rebutted key concerns and suggestions without sufficiently changing the text of the paper. If they cannot add necessary data to this current paper, then they need to change the text, such that they do not need to rely on these findings to support their study design or interpretation of the data they are presenting. In other words, they need to remove every instance of relying on information that is in an as yet unpublished ‘other’ paper, or is for some other reason not published and not provided in this paper. For example, throughout the paper, the authors refer to doses/concentrations of BaP as sub-toxic, but provide no empirical evidence to support this. Their rebuttal explains that these data are in another paper that they wish to submit elsewhere. That is fine, but they therefore need to re-write sentences throughout this paper to remove these claims. They also refer to the concentrations used as environmentally relevant. This may or may not be the case, my point is that the need to substantiate this, and other claims, in this paper (e.g., reference prior studies that support the veracity of this statement).
Several other points raised in the initial review received extensive explanations in the authors’ rebuttal letter, but only modest changes were made to the paper. As one example, it was pointed out that the authors over-interpret some of their data, for instance by mentioning trends (a statistically meaningful word that also implies that studies have been independently replicated, and the pattern of change is consistently observed). This text was not revised, and needs to be.
There are other instances in which concerns raised were rebutted but limited or no change made to the paper. For example, the authors continue to interpret some of the data (e.g., bacterial uptake, cytokines) with cell-type specific conclusions; however, many of the assays and approaches do not provide cell-type specific data. To be clear, the concern is not that these assays are flawed or inappropriate. But, they do not provide the information reflected in the conclusion. Therefore, the text throughout the paper needs to be revised to remove statements that are not supported by the approaches used and/or the data obtained.
Several examples are provided in the spirit of giving constructive and helpful examples of how they could modify the text of the paper to make this interesting set of studies even more compelling and to ensure that their conclusions are robustly supported and predicated on information that is transparently presented. These are examples only, and should not be misconstrued as a complete summary of edits needed throughout the paper.
Abstract: Lines 37-39: please revise because this sentence implies that you infected AHRKO mice with S.E., which was not done.
Abstract: Lines 39-40: please revise, as your experiments did not show that the higher Ab levels in BaP treated mice are due to bacteria persisting longer. This is an interesting speculation, and would be appropriate to consider upon in the Discussion, but it is not suitable to present as a result in the Abstract.
Introduction: The Introduction is overly long. The content is not uninteresting, but seems excessive—and has a lot of details that don’t seem to be necessary to Introduce the study presented in your paper. Some of the information in the Introduction could be incorporated into the Discussion.
Results: The paper would be stronger rather than weaker if you limited interpretations of data to what information the approaches used provide.
Results: The use of AHR KO mice is an important attribute of the work in this paper. The authors presented, in their rebuttal, a lot of information as to why they did not infect AHR KO mice with SE and perform more extensive assessments. Given how difficult these mice are to work with, their frustration using with them is understandable. Yet, this section of the paper needs to be revised to be clearer and more circumspect. For example, sometimes the sentences in the text do not seem to align with the data in the figure (e.g., panel 1 showing IL-10 levels in peritoneal cavity cells—BaP is black bars, control is grey bars; yet, the text (lines 10-20-1021) mentions that IL-10 was significantly higher following BaP exposure). The sentence in lines 1022-1024 is not supported by the data. This sentence contains an interesting speculation (so could mention in the Discussion), but the data in this figure do not robustly demonstrate that IL-10 production by immune cells is AHR-dependent. More rigorous research needs to be performed to support this. If this is more than the authors wish to undertake, which is understandable, then they need to revise the text to remove statements such as this (this is but one example of this issue). Similarly, the sentence in lines 1036-37 needs to be revised because it’s meaning is confusing. What I see is that peritoneal cells from hk-SE treated mice given BaP produced less TNF, and this decrease was not observed in AHR KO mice. This suggests that this effect of BaP is AHR-mediated, which the authors state. What is confusing is the ‘but not BaP dependent.’
Results: The authors mention in their rebuttal removal of several figures and tables related to gene expression analyses and in vitro studies, as recommended. However, it seems that these are still in the revised paper.
Discussion: The Discussion is extremely long, restates a lot of the results. The authors are encouraged to consider reducing the text describing the results and enriching the Discussion section by integrating their findings with current knowledge. As mentioned previously, there is quite an extensive literature on the immunotoxicity of BaP, published by different groups of authors, yet this work remains absent in the current paper. Inclusion is necessary.
Discussion: As in other parts of the paper, please remove or revise unsubstantiated statements, such as referring to the doses of BaP used as being non-toxic.
